# Multi-level storylines for participatory modeling — involving marginalized communities in Tz'olöj Ya', Mayan Guatemala

Jessica A. Bou Nassar[1], Julien J. Malard[1], Jan F. Adamowski[1], Marco Ramírez Ramírez[2], Wietske Medema[1], Héctor Tuy[2]

[1]Dept. Bioresource Engineering, McGill University, 21111 Lakeshore Road, Sainte-Anne-de-Bellevue, QC, H9X 3V9, Canada
[2]IARNA, Universidad Rafael Landívar, Vista Hermosa III, Campus Central, Zona 16, Edificio Q, Oficina Q-101, Ciudad de Guatemala, Guatemala

*Correspondence to:* Jessica A. Bou Nassar (jessica.bounassar@mail.mcgill.ca)

**Abstract.** Unconventional sources of data that enhance our understanding of internal interactions between socio-economic and hydrological processes are central to modeling human-water systems. Participatory modeling (PM) departs from conventional modeling tools by informing and conceptualizing human-water systems through stakeholder engagement. However, the implementation of many PM processes remains biased, particularly in regions where marginalized communities are present. Many PM processes are not cognizant of differentiation and diversity within a society and tend to treat communities as homogeneous units with similar capabilities, needs, and interests. This undifferentiation leads to the exclusion of key actors, many of whom are associated with marginalized communities. In this study, a participatory model-building framework (PMBF), aiming to ensure the inclusiveness of marginalized stakeholders - who (1) have low literacy, (2) are comparatively powerless, and/or (3) are associated with a marginalized language - in participatory modeling is proposed. The adopted approach employs interdisciplinary storylines to inform and conceptualize human-water systems. The suggested method is underpinned by the Multi-level Perspective (MLP) framework, which was developed by Geels et al. (2002) to conceptualize socio-technical transitions and modified in this study to accommodate the development of interdisciplinary storylines. A case study was conducted in Atitlán Basin, Guatemala, to understand the relationships that govern the lake's cultural eutrophication problem. This research integrated key stakeholders from the indigenous Mayan community, associated with diverse literacy ranges, and emerging from three different marginalized linguistic backgrounds (Kaqchikel, Tz'utujil, and K'iche'), in the PM activity. The proposed approach facilitated the participation of marginalized stakeholders. Moreover, it (1) helped develop an understanding of mechanisms governing the eutrophication of the lake, (2) initiated a dialogue between Indigenous Peoples and non-Indigenous stakeholders, and (3) extracted potential solutions targeting the system's leverage points. The participatory model-building activity generated three submodules: (1) agriculture, (2) tourism, and (3) environmental awareness. Each submodule contained socioculturally specific mechanisms associated with nutrient discharge to Lake Atitlán. The delineation of such nuanced relationships helps develop well-targeted policies and best management practices (BMPs). Additionally, the suggested process helped decrease the impact of power imbalances in water resources management and empowered community-based decision-making.

## 1 Introduction

Cultural eutrophication and associated algal blooms have become prevalent in freshwater ecosystems worldwide (Smith and Schindler, 2009). Anthropogenic activities (e.g. agricultural, industrial, and residential) have exacerbated the trophic states of lakes by increasing the associated discharge of point-source and nonpoint-source limiting nutrients (Schindler, 1974). Such water quality problems are challenging to solve as they are characterized by the complex interactions between biophysical and socio-economic dimensions (van Bruggen et al., 2019; Gunda et al., 2018). Deterioration of lake ecosystems due to cultural eutrophication is especially magnified in developing countries, where governing bodies tend to be more tolerant of practices contributing to aquatic nutrient enrichment

(Nixon, 1995; Withers and Haygarth, 2007). To address problematic human-water interactions in developing countries, the bottom-up development of management practices and policies with stakeholders is crucial (Perrone et al., 2020).

Conventional modeling tools (e.g. physically based models) are often ill-suited for addressing the challenges mentioned above, since they fail to endogenously incorporate socio-economic processes when addressing hydrological problems (Inam et al., 2017a; Malard et al., 2017). They are also complex, lack transparency, and are often incompatible with participatory methods. Consequently, they reinforce expert-oriented and externally imposed opinions, which tend to lack situated knowledge (Cooke and Kothari, 2001; Inam et al., 2015). As such, water resources management requires transformative interdisciplinary methods, such as participatory modeling of human-water systems, to better capture local realities and improve understanding of the socio-economic factors impacting water-related problems (van Bruggen et al., 2019; Inam et al., 2015).

Systems thinking is a powerful tool for participatory modeling (PM) (Inam et al., 2017a). Systems thinking can capture socio-economic processes elicited from stakeholders and can accommodate nonlinearity and multi-causality. It can also delineate iterative bi-directional feedbacks embedded in human-water systems (Prodanovic and Simonovic, 2010). The identification of such feedbacks is important to better inform and conceptualize human-water systems. Furthermore, systems thinking can be accompanied by visual aids, generating more comprehensible and stakeholder-friendly models (Alcamo, 2008). As a result, systems thinking can accommodate stakeholder participation and enhance model development with situated knowledge.

PM can incorporate stakeholders in decision-making through its departure from conventional model building, packaging, and dissemination processes (Voinov et al., 2016). However, the implementation of such processes - particularly in regions with marginalized communities (i.e., less literate, comparatively powerless, or associated with marginalized languages) - is challenging. Many PM processes do not focus on diversity and differentiation within a society and tend to treat communities as homogeneous units with similar needs, capabilities, and interests (Bohensky and Maru, 2011; Guijt and Shah, 1998). Undifferentiated treatment in PM can lead to the exclusion of key actors, especially marginalized communities. As such, three issues are raised. First, many PM activities require professional skills and expertise, thereby preventing the involvement of less literate stakeholders (Inam et al., 2015; Maynard and Jacobson, 2017). Second, many participatory methods usually overlook group dynamics, yielding participatory decisions that reinforce the interests of those in power ((Cooke and Kothari, 2001; Eker et al., 2018). Third, participatory model-building processes might fail to recognize integrated participation in multilingual regions, which can further marginalize Indigenous languages (e.g. Hassanzadeh et al. (2019)).

One of the broad aims of many participatory approaches is to increase the involvement of socially and economically marginalized communities in making decisions that impact them and are impacted by them (Guijt and Shah, 1998; Izurieta et al., 2011). This is necessary for several reasons. First, marginalized stakeholders play vital roles in water resources management. Thus, they can be primary contributors to model-building activities and finding appropriate solutions for the problems being explored (Colfer and Dudley, 2011; Figueiredo and Perkins, 2013). For example, many marginalized communities are involved in agriculture and aquaculture and have sufficient experience to determine the practices that could be successfully integrated into everyday practices and adopted by corresponding actors (Hassanzadeh et al., 2019). Second, marginalized communities are often the most vulnerable to environmental change, such as water quality degradation of freshwater ecosystems. Therefore, these communities should have the right to participate in decisions that affect their environment, lives, and well-being (Evans, 2006). Third, inclusive participation in policymaking can facilitate sustainable management. While politicians and businesses are often interested in short-term benefits, communities tend to focus on long-term solutions that ensure the availability of water resources for future generations (Colfer, 2005). Finally, earlier research established that interactions between different participants with diverse backgrounds and perspectives are crucial in participatory processes, increasing creativity and producing new insights (Funtowicz and Ravetz, 1993; Martins et al., 2018; Webler, 1995). Therefore, to align the objectives of PM with the concerns outlined above, approaches that ensure the inclusion of marginalized stakeholders in such processes are needed.

Some participatory methods supporting the inclusion of marginalized stakeholders in PM and data collection processes have been suggested. For example, the Rich Pictures approach uses pictures and symbols in an

unstructured way to capture flows of information, communication, and human activity (Berg and Pooley, 2013). The method aims to accommodate participatory activities in culturally diverse, less literate, and multilingual communities (Berg and Pooley, 2013; Colfer and Dudley, 2011; Voinov et al., 2018). However, the use of symbolism and pictures yields ambiguity and can be misinterpreted (Lewis, 1992).  Therefore, this method is not necessarily well-suited for portraying the complexity of human-water interactions.

Spatial mapping has also been used for facilitating the inclusion of stakeholders, with little to no literacy, in participatory activities. This approach allows local stakeholders to (1) generate maps depicting information and knowledge – the 'where' and 'how' – associated with a problem, and (2) reveal their perceptions of that problem. Participatory spatial mapping has been useful for triggering discussions between stakeholders but is not suitable for exploring future scenarios. Although the method has been successfully applied in the data collection process of
participatory research (Rambaldi et al., 2007; Reilly et al., 2018), it is not well-suited for the conceptualization of human-water systems, as they encompass complex interactions between spatially and temporally distant components and non-spatial variables  (Di Baldassarre et al., 2017; Forrester, 1969).

     Additionally, facilitation techniques, such as 'Fish Bowl' – an activity that allows each participant a brief period to express views on the investigated issue – or 'Line on the Floor' – an activity where a line on the floor
represents a boundary between two categories of stakeholders with different opinions – was suggested by Colfer and Dudley (2011) to include less literate stakeholders in participatory activities. This genre of activities can only be conducted in group sessions, and there is the problem of the potential effects of unhealthy group dynamics. Stakeholders are often more likely to engage in individual rather than group sessions and communicate openly when alone (Burgin et al., 2013; Videira et al., 2009). Moreover, these methods could have challenges in eliciting the
detailed stakeholder perceptions that are required by PM processes.

     Another approach, stakeholder created causal loop diagrams (CLDs), contain variables connected by links indicating causal relationships. Causal loop diagrams have been previously applied in water resources management (e.g. Hassanzadeh et al., 2019; Stave, 2003) In many cases, their construction required reading and writing skills (e.g. Inam et al., 2015, 2017b; Perrone et al., 2020) or technical skills (e.g. Mavrommati et al., 2014; Tidwell et al.,
2004). This can pose challenges when involving less literate stakeholders in participatory model-building activities. In some studies, causal loop diagrams were extracted from interviews or focus group discussions and processed by researchers, ex-post (e.g. Enteshari et al., 2020; Giordano et al., 2020; Pham et al., 2020; Santoro et al., 2019). There are two challenges related to this: (1) it increases the risk of researchers' influences on the model and (2) it might yield ambiguous statements, prone to misinterpretation (Kim and Andersen, 2012). Both are especially critical in the
context of marginalized communities, where perspectives of less-powerful stakeholders often tend to be lost or disregarded (Butler and Adamowski, 2015; Cooke and Kothari, 2001).

     The primary focus of this research is the implementation of a participatory method that facilitates the inclusion of traditionally marginalized stakeholders, who are (1) less literate, (2) relatively powerless, and/or (3) associated with marginalized languages, in modeling human-water systems. The method suggests an extension to
CLD-building to facilitate inclusion. The integration of storylines with causal loop diagrams through the Multi-level Perspective (MLP) framework is proposed to enhance the involvement of marginalized stakeholders in PM processes. The MLP framework was initially developed by Geels (2002) to conceptualize socio-technical transitions and explains developments in and interactions between three levels: landscape, regime, and niche (elaborated in subsequent sections). The framework was adjusted in this study to accommodate the interdisciplinary development of storylines.
The objectives of the study are to:

1. Propose a conceptual framework for building multi-level storylines that (1) is inclusive by design and (2) can inform and conceptualize human-water systems, by adjusting the MLP framework.
2. Suggest a framework for the implementation of the storyline construction process that (1) facilitates the participation of less literate stakeholders, (2) reduces unhealthy power dynamics, (3) accommodates a
multilingual context, and (4) makes use of the system's leverage points to select best management practices (BMPs) and policies.

3. Evaluate the validity of the process with respect to its ability to (1) incorporate effective participation of marginalized stakeholders, (2) induce a dialogue, (3) integrate diverse perspectives, (4) facilitate model-conceptualization, and (5) produce descriptions of relevant human-water feedbacks.

A case study was carried out in the Atitlán Basin, Guatemala, which integrated stakeholders from the Indigenous Mayan community into the proposed participatory model-building process to fulfill the third objective. This case study was selected since it incorporates relatively powerless stakeholders, associated with diverse literacy ranges, and belonging to three different marginalized linguistic backgrounds: Kaqchikel, Tz'utujil, and K'iche'. The study applied the proposed storyline development framework to investigate the relationships that govern the

eutrophication problem in Lake Atitlán from a holistic community-based perspective and empower community-based decision-making. The remainder of the paper is structured as follows: Section 2 discusses the conceptual framework for multi-level storyline development. Section 3 provides background information for the case study. Section 4 provides a stepwise approach for implementation of the multi-level storyline development framework. The results of the implementation of the process in the Atitlan Basin are presented in Section 5. Section 6 evaluates the

results and discusses them from the perspective of human-water feedbacks, and Section 7 concludes the study.

## 2    Conceptual Framework

In this section, the building blocks of the method – storytelling and the MLP framework- are discussed. An argument for using storyline development to facilitate the inclusion of marginalized stakeholders in conceptualizing human-water systems is presented. Finally, the conceptual framework for the development of multi-level storylines

is elaborated.

### 2.1    Storytelling

Storytelling techniques are a way to visualize and describe conditions using oral or textual narration, to provide information and insight (Hazeleger et al., 2015; Moezzi et al., 2017; Zscheischler et al., 2018). This method helps people from different domains, and professional and sociocultural backgrounds better understand different

perspectives since it provides leeway for elaboration and does not restrict the communicator with a technical approach. The storytelling approach is suggested for helping to solve water resources problems where (1) cross-dimensional collaboration across different fields and entities (e.g. agriculture, government, and academia) is necessary to ensure a holistic understanding of the problem, policy outcomes, and potential risks (Thaler and Levin-Keitel, 2016; Treuer et al., 2017), and (2) the interconnectedness of different domains transcends hydrological

systems and involves the implementation of generated decisions (Haeffner et al., 2018; Hassanzadeh et al., 2019). Storytelling can also help accommodate the participation of marginalized stakeholders. Since storylines are usually communicated verbally, the process requires neither reading nor writing skills and, therefore, is compatible with the involvement of less literate stakeholders in participatory activities (Colfer and Dudley, 2011). The method allows participants to use anecdotes and metaphors to describe their observations. This is useful in the context of

less literate stakeholders or non-modelers who might not be able to explicitly portray their observations in a technical manner. Also, it can be carried out either in individual sessions – to reduce unhealthy power dynamics (Butler and Adamowski, 2015) – or in group sessions – which is necessary when discussions between participants of different perspectives are required (van Bruggen et al., 2019; Evans, 2006). Storytelling allows for the portrayal of the studied issue in detail and with reduced ambiguity since it encourages participants to elaborate on their

descriptions of conditions. The elicited storylines provide researchers with knowledge and information while also aiding in model conceptualization, characterization of future scenarios, and evaluation of modeling results (Alcamo, 2008; Trutnevyte et al., 2014). Due to the flexibility of the storytelling process, storylines can also consider nonlinearities, multi-causality, and complex causal links (Arico et al., 2001). Therefore, they are well-suited for helping to inform and conceptualize systems models. Data sources that can enhance understanding of and capture

human-water feedbacks are needed for the development of holistic, participatory models that represent complex interactions between hydrological and socio-economic variables (Mount et al., 2016). The highly descriptive and

flexible nature of storytelling helps capture the empirically observed complexity associated with such phenomena (Leong, 2018).

Storylines have been used by many researchers to complement models (Arico et al., 2001; Booth et al., 2016; Trutnevyte et al., 2014). Guhathakurta (2002) stated that storylines underpin models as a means of reconstructing and investigating stories. In addition, Trutnevyte et al. (2014) stated that the iteration between storylines and model results could correct over or underestimations depicted by either. Nevertheless, the incorporation of participatory storytelling techniques in environmental modeling and resource management has been limited (Arico et al., 2001; Carpenter et al., 2015; Cobb and Thompson, 2012; Delmotte et al., 2017; Treuer et al., 2017). Methods guiding participatory storytelling have focused on conducting interviews with stakeholders, carrying out collective workshops, developing appropriate focal questions, and iterating between model results and stakeholders (Arnell et al., 2004; Booth et al., 2016; Cobb and Thompson, 2012; Foran et al., 2013). However, these storytelling approaches have been specifically designed to inform conventional models (such as physically based models) and are not necessarily well-suited for systems modeling.

The storyline construction processes used in the above-listed studies start with requiring stakeholders to state the most significant or uncertain drivers that are expected to shape the future trajectory of the modelled problem. Hence, those techniques usually frame the resulting models with selected drivers of change, which are the initiators for the storyline development process. For example, Delmotte et al. (2017) held a workshop in which drivers of change were identified and ranked by stakeholders, and the two most prominent drivers were selected: (1) climate change and (2) economic conditions for rice cultivation. Then, a two-dimensional matrix was built, depicting the extrema of the driver states: (1) low and high climate change impacts (x-axis) and (2) favourable and unfavourable economic conditions for rice cultivation (y-axis). This matrix was then used to instigate four plausible storylines from each of its quadrants. This concept is dominant in storyline construction processes and is convenient for informing physically based models, in which driving forces are only interacting exogenously with other modelled variables. However, in systems thinking and modeling approaches, prior to considering driving forces, interactions between diverse components that cause and reinforce the problem are required. In other words, the problem, as-is, is created by eliciting the relationships essential to its continuance. Therefore, the problem's triggers are not considered as external 'drivers' imposed on the system but rather internally acting and reacting within the modelled structure (Forrester, 1969).

Additionally, the key mechanism for exploring plausible futures or scenarios using systems thinking is through adding a component (or more) to the system, adjusting a certain trend of a component (or more) of the system, or both. Unlike conventional scenarios produced by physically based models, which are shaped by external drivers of change, scenarios derived from systems thinking are characterized by components that are endogenously interacting within the system. Therefore, the unique nature and structure of systems models require a different storytelling technique that produces storylines capable of informing and conceptualizing the founding relationships of the model and characterizing future scenarios using internal model variables.

The notion of coupling storylines with systems thinking has been previously suggested (Geum et al., 2014; Mallampalli et al., 2016; Olabisi et al., 2010). Mallampalli et al. (2016) highlighted the suitability of systems modeling for quantifying narratives but did not elaborate on associated storyline construction methods. Olabisi et al. (2010) developed different socio-ecological scenario storylines with stakeholders; each storyline described a plausible future corresponding to the year 2050 in Minnesota, driven by certain elements (e.g., natural, social, political) and associated trends. The authors then constructed several systems models underpinned by a scenario storyline, containing relationships that only represented the year 2050. The model results were only used to evaluate the consistency of scenario storylines. The storyline construction process used by Olabisi et al. (2010), and elsewhere (Geum et al., 2014), was initiated by identifying driving forces and outcomes of alternative futures, excluding systems thinking from that phase of the process. In other words, systems thinking was not used as a tool to explore possible future states of the modelled system; systems modeling was used to simulate pre-built and previously conceptualized future scenarios.

Although this approach is useful for providing visions of alternative futures, it is not necessarily well-suited for designing decision-support tools, testing policies and BMPs, and generating policy-based scenarios for water

resources management. This method does not make use of one of the key advantages of systems thinking: the ability to expose leverage points. A leverage point is a position in a system where a minimal shift generates a major change in the system's functioning (Meadows, 1999). The majority of leverage points cannot be identified intuitively. Even if a leverage point is delineated intuitively, it is often misused, leading to unintended system behavior. In other words, relationships governing leverage points are counterintuitive (Forrester, 1971). Therefore, the identification of leverage points requires a thorough exploration of the modelled system as-is (prior to projecting it) and an understanding of its components and relationships. In return, the detection of leverage points aids decision-making by highlighting where a policy or BMP could be assigned to yield a transformative change in the system's state. In this context, BMP or policy-based scenarios should be suggested and generated in the later phases of the modeling process and not at the initial phase. Hence, this study presents a framework for the construction of interdisciplinary storylines that aim to (1) inform and conceptualize models using systems thinking and (2) make use of leverage points to empower decision-making.

## 2.2     Multi-level Perspective (MLP) Framework

The MLP framework (discussed in detail elsewhere: Geels and Kemp, 2000; Geels, 2002; Kemp et al., 2001), was developed for the analysis and description of socio-technical transitions (Timpe and Scheepers, 2003). This framework has been widely adopted for depicting transitions in the electricity sector (Foxon et al., 2010, 2013; Moallemi et al., 2017; Moallemi and Malekpour, 2018). The framework has also been used to describe transitions in water governance (e.g. Daniell et al., 2014; Orr et al., 2016; Xu et al., 2016).

The MLP framework was established to explain the development of technology from interactions occurring within and between different levels: landscape, regime, and niche. The landscape represents the Macro-level, which contains external factors that bind and contextualize transition trajectories. It involves a set of heterogeneous factors (e.g., social structure and political coalitions) and defines the environment for developments and corresponding interactions. The regime delineates the Meso-level, reflecting the stability of existing developments in technology. It outlines the rules that restrain activities within communities, setting the environment for the occurrence of socio-technical transitions. The niche depicts the Micro-level, accounting for the radical innovations which are not yet part of the dominant regime (Geels, 2002). The relationship between the three concepts is a nested hierarchy, implying that landscapes contain regimes and regimes contain niches. Therefore, niches emerge within the context of the prevailing regimes and corresponding landscapes, according to associated rules and capacities. The prevalent regimes and landscapes strongly influence the emergence of niches. This highlights the significance of the alignment of developments at the three levels, by which existing arrangements play a significant role in shaping innovations at the niche level and in determining whether associated radical innovations will yield a shift in the dominant regimes (Kemp et al., 2001; Mylan et al., 2019).

The MLP framework has not been used, in the context of systems thinking, for the development of storylines that aim to inform and conceptualize models and, therefore, is modified in Sect. 2.3 in this study to accommodate the latter. This study builds on three concepts of the MLP framework: (1) the three levels, (1) the nested hierarchy of levels, and (3) the recognition that existing arrangements play a central role in shaping future developments of the system. In this paper, the three levels are referred to as Macro-level, Meso-level, and Micro-level, instead of landscape, regime, and niche, respectively.

## 2.3     Integrated Approach: Multi-level Storylines

Storylines developed to conceptualize a systems model should inform (1) the boundaries of the system representing the problem, (2) the components and interactions that make up the system (contained within the boundaries), and (3) the desired BMPs and policies within the context of the modelled problem– ideally targeting leverage points. The construction of conceptual models using storytelling is, therefore, underpinned by the integration of storylines developed at three levels: Macro, Meso, and Micro (Geels, 2002). The relationship between the three levels is depicted as a nested hierarchy. Meso-level storylines are within the scope of Macro-level

storylines and informed by them, and Micro-level storylines are within the scope of Meso-level storylines and informed by them. Understanding and structuring the constituents of the storylines from stakeholders at each level is required to facilitate storytelling and model conceptualization processes.

The Macro-level storyline sets the gradient for all plausible present and future outcomes produced by the model. It contains historical influences, social and geographical contexts, the problem definition, and the assigned time horizon (Convertino et al., 2013; Inam et al., 2015). Hence, it provides the boundaries and scale of the modelled system, which are essential for initiating the model's conceptualization and informing the Meso-level storyline. The Meso-level storyline portrays the modelled problem's state, which is yielded by dynamic interactions between the components of the problem, contained by system boundaries. It is made up of the causes and

consequences of the problem, and the relationships and feedbacks between them. The storyline is designed to depict the problem and the corresponding state as-is. Translating the Macro-level and Meso-level storylines into a CLD allows for the exploration of some of the system's leverage points. Subsequently, this informs the Micro-level storylines, which encompass BMPs or policies and corresponding outcomes within the context of the modelled problem. For effective policy selection, candidate policies (policies that are deemed suitable by several stakeholders)

contained by the Micro-level storylines should target leverage points and undesired outcomes. Policies can either (1) restructure or reconfigure the system, or (2) strengthen or weaken dynamics already embedded within it. The emergence and simulation of certain BMPs or policies then depict the starting point of the corresponding policy-based scenario. However, the changes induced by and the outcomes of the simulated BMPs or policies are underpinned by, and occur, within an existing system. Therefore, the exploration of the dominant system's

arrangements that shape and influence plausible future developments is crucial prior to constructing Micro-level storylines. Hence, having a holistic view of the system allows for the establishment of policies and BMPs that target long-term transformation of the system's problematic state, rather than short-term remedies (Forrester, 1969). The components of storylines associated with each level are displayed in Fig. 1. The figure shows that policies contained by Micro-level storylines should be aligned with depicted leverage points or undesired outcomes. It also displays

multiple policy options for a single selected point.

Multi-level storylines can be used in parallel with CLD-building to facilitate more inclusive stakeholder participation. Storylines provide an opportunity for stakeholders to describe their observations, using, for instance, anecdotes and metaphors. This is particularly useful in the presence of less literate or "non-expert" stakeholders who might not be comfortable with the technical aspects of CLD-building and might not explicitly place their

observations in the context of variables and links. Additionally, disseminating and communicating results in the form of storylines is more suitable for an audience of non-modelers, especially in the context of marginalized communities that include stakeholders who might not be comfortable with deciphering CLDs. Moreover, the method is explicitly and systemically designed to dynamically translate from storylines to CLDs and vice versa, which makes (1) stakeholders' statements less prone to misinterpretation and (2) the process less susceptible to researchers'

influences, compared to other CLD-building processes that require ex-post extraction of CLDs from interviews or focus group discussions (Giordano et al., 2020; Pham et al., 2020). This facilitates the conservation of stakeholders' views.


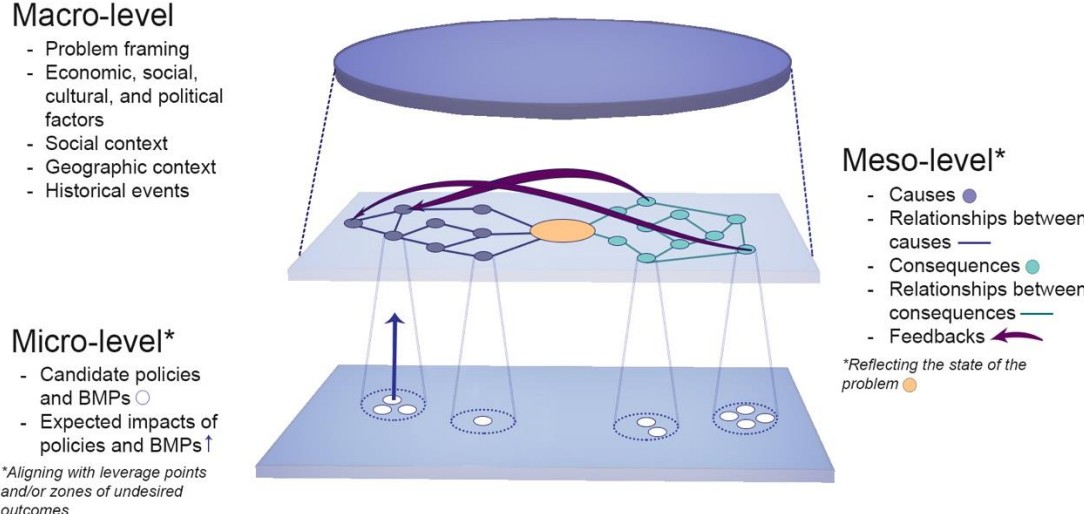

**Figure 1: Components of storylines at three level: Macro, Meso, and Micro (modified from Geels et al. (2002) to accommodate multi-level level storyline development)**

## 3   Case Study

Lake Atitlán is the deepest lake in Central America, with an average depth of 220 m and a maximum depth of 341 m. Located in the southwestern region of Guatemala, it is a highland, endorheic lake formed in a collapsed caldera. The lake's surface area is 137 km$^2$, while the Lake Atitlán watershed is 541 km$^2$ (Fig. 2) (Ferráns et al., 2018; Newhall, 1987). Lake Atitlán is a warm monomictic lake that experiences two main seasons: (1) dry from November to April and (2) wet from May to October (Weiss, 1971). More than 50% of the watershed consists of steep slopes (Komárek et al., 2013).

The Atitlán Basin contains numerous point- and nonpoint-sources of nutrient pollution. The most prominent are agricultural runoff, untreated wastewater, and eroded soils (Weisman et al., 2018). For the past several decades, increased development of the area, coupled with poor environmental management practices and policies, has yielded a surge in nutrient loading to the lake. This ongoing process of cultural eutrophication has recently shifted the lake's state from oligotrophic to mesotrophic (Komárková et al., 2011). Lake Atitlán experienced a very large cyanobacteria bloom covering 40% of its surface in October 2009 (Komárek et al., 2013).

The Atitlán Basin encompasses 15 municipalities and approximately 300,000 people (INE, 2018). Forests and agricultural areas cover more than 70% of the watershed (Komárková et al., 2011). Agriculture, aquaculture, and tourism are the dominant economic sectors in the region (Ferráns et al., 2018) . The Atitlán Basin is home to three Mayan communities: Kaqchikel, Tz'utujil, and K'iche'. The marginalization of these communities is magnified at institutional levels (national and local) and in education systems, where associated Indigenous languages are seldom acknowledged. These Indigenous communities are dependent upon the lake and value it economically, socially, and spiritually. The cyanobacterial blooms in 2009 hindered drinking, fishing, and leisure activities, which are crucial for the lives of Indigenous communities in the vicinity of Lake Atitlán.

In 2018, government authorities endorsed a proposed project (referred to as the 'Mega-collector') to enhance the lake's water quality. The project involves building large, centralized infrastructure to collect wastewater from all the towns encircling the lake and transporting it to a treatment plant outside the watershed. The wastewater would then be treated and used by agroindustrial farms for irrigation. According to discussions with stakeholder, some Indigenous communities have raised objections for several reasons. First, they are concerned with the reallocation of the watershed's water resources due to associated implications on the basin's water shortage problem

and the inequitable distribution of benefits. Second, they emphasize that such a large-scale project would have very negative impacts on the lake's ecosystem and biodiversity. Third, since the basin encompasses multiple seismic faults, some Indigenous communities question the resilience of large infrastructure in an earthquake-prone zone.

Fourth, they highlight that the project would not solve the eutrophication problem definitively since it disregards other contributing factors such as agricultural runoff and soil erosion.

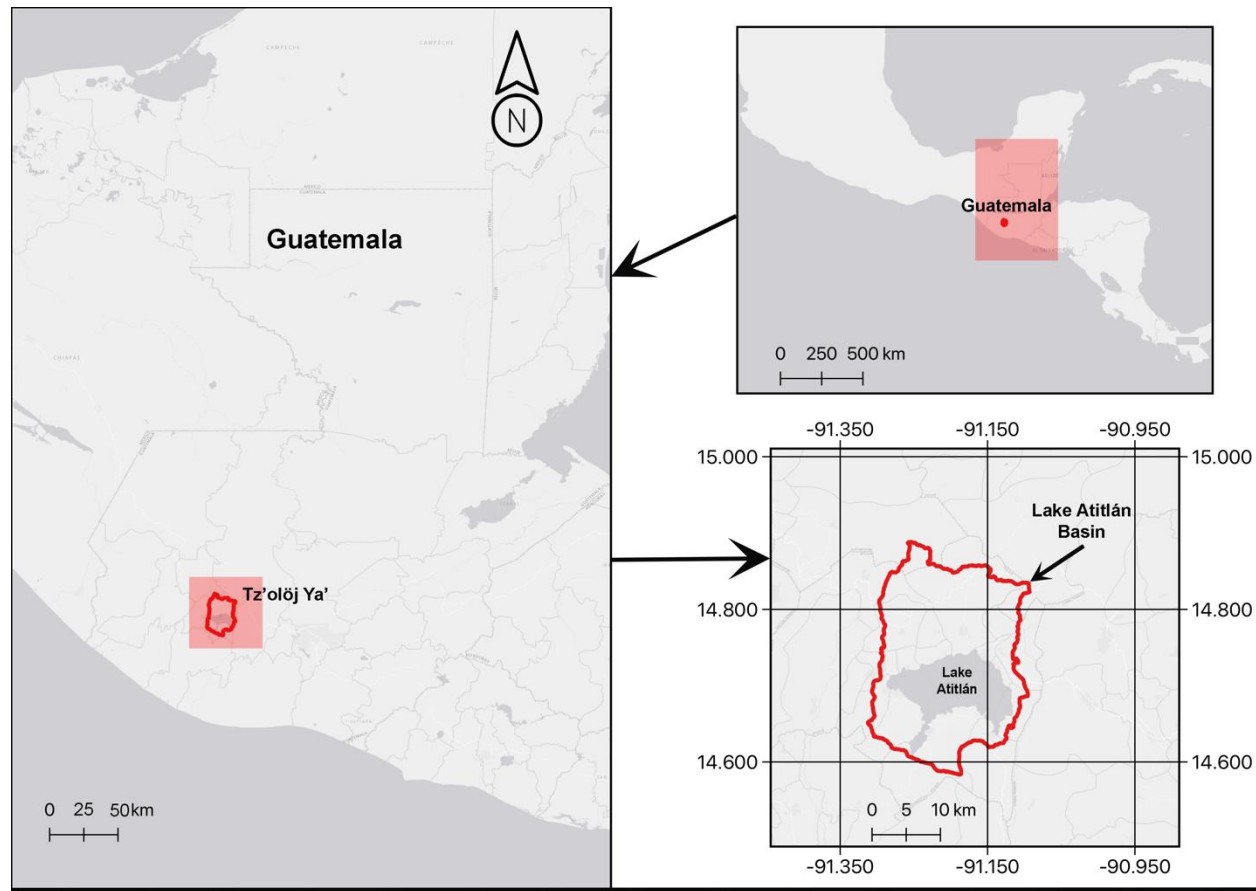

Figure 2: Location of the study area in Guatemala. Created in QGIS using Esri (2009).

## 4 Methodology

The proposed storyline development process takes PM activities in multilingual contexts into account. Therefore, prior to initiating the process, a multilingual guidance team is developed. The team consists of experts and
365 organizers. At least one person with a good command of each language included in the project and the corresponding region is present in the team.

### 4.1 Stage 1: Macro-level Storylines

1. Identifying researcher participants: Researcher participants (stakeholders from local institutions researching in the study area) are selected to construct Macro-level storylines. It is important to select researcher

participants from different professional and sociocultural backgrounds and who identify as belonging to marginalized groups, to construct a holistic view of the problem.

2. Developing a focus group with primary stakeholders: A focus group is created where the guidance team provide language translations between stakeholders. The purpose of the focus group is to:

        a. Frame the problem: the problem should not be defined too narrowly as it will take its definite

shape after subsequent interviews with the complete group of participating stakeholders (Arico et al., 2001)

        b. Contextualize the study system by delineating dominant economic sectors, power imbalances, cultural diversity, and the region's political culture, among others (Mostert, 2018)

        c. Set the social and geographic contexts of the model

380         d. Outline historical events that have influenced the problem (Foran et al., 2013)

Stakeholders share information in narrative form. The guidance team leads the discussion to obtain the information required to build the model's Macro-level storylines. However, they refrain from restraining participants' ideas or opinions. They also ensure that marginalized communities are discussed. Narratives are recorded in writing. This step aids the guidance team in enhancing situated knowledge and recognizing

their positionality in the model-building process, while also providing the context for the Meso-level and Micro-level storylines.

4.2     **Stage 2: Developing Meso-level Storylines**

1. Performing a stakeholder analysis: The Macro-level storyline informs the stakeholder analysis process. The primary stakeholders selected in Stage 1, along with members of the guidance team, brainstorm to identify

other relevant stakeholders (Calvert, 1995; Vos and Achterkamp, 2006). The guidance team explicitly delineates stakeholders representing the different dimensions (economic, social, cultural, and political), mentioned in the Macro-level storyline. The team actively seeks individuals and organizations that are associated with marginalized communities.

2. Stakeholders participating in the model-building process are then grouped according to their roles (i.e.

decision-makers, users, implementers, and experts) and attributes (i.e. power, urgency, interest, and legitimacy) and selected to ensure that at least one person representing each role and attribute is included (Freeman, 2010; Inam et al., 2015; Mitchell et al., 1997).

3. Conducting individual semi-structured interviews with stakeholders: The guidance team prepares focal questions to direct the construction of the Meso-level storylines and carries out individual semi-structured

interviews with all participants. Interviews are conducted individually to minimize the influence of power dynamics on the model-building process (Ayrton, 2018; Butler and Adamowski, 2015; Colfer and Dudley, 2011; Inam et al., 2015). Semi-structured interviews are used since they allow interviewees to speak more freely (Ayrton, 2018; Elsawah et al., 2015; Voinov et al., 2018). Since some stakeholders might not be comfortable with their narratives being recorded, interviewers only take notes of the interview (Elsawah et

al., 2015; Strauss and Corbin, 1990). Also, participants are asked to use linguistic statements that reflect qualitative knowledge (e.g. when X increases, Y decreases) to extract storylines that are meant to conceptualize systems models (Alcamo, 2008). The role of the interviewer is to extract phrases containing indicators that can be estimated.  When an interviewee states an ambiguous concept, the interviewer asks the interviewee to explain more, until a tangible relationship between definite variables is identified. The

steps of the interview process are elaborated below.

        a. A focal question is formulated by the guidance team to elicit direct and indirect causes of the problem (Arico et al., 2001).  For example: what are the underlying causes of the investigated problem? Stakeholders are asked to respond to the focal question in a set of coherent statements, building storylines.

b. The single-driving force method (Fig.3) is used as a starting point to elicit direct and indirect consequences yielded by the problem. As per the field guide established by Evans (2006),

narratives can be elicited using the single-driving force technique by asking questions such as: (1) what happens if the problem is reinforced? (2) What happens if the problem is diminished? (3) What happens next? (4) What are the consequences of that? The chain of questions derived from the single-driving force method is prolonged to elicit feedback effects of consequences on pre-stated causes.

**Figure 3: The single-driving force method**

4.  Each of the extracted narratives is translated into an individual CLD by the guidance team. A CLD is made up of variables and causal links between them (Fig. 4). The sign corresponding to each link indicates the type of relationship between the two variables: (+) indicates a positive causal relationship (i.e., when the causative variable increases, the effect increases and when it decreases, the effect decreases), while (-) implies a negative causal relationship (i.e., when the causative variable increases, the effect decreases and when it increases, the effect decreases). Two types of feedback loops exist: balancing (Fig. 4 (a)) and reinforcing (Fig. 4 (b)) (refer to Inam et al., 2015). The semi-structured interview (elaborated in the previous step) is designed to elicit narratives containing identifiable causes, consequences, and feedbacks. Therefore, this step requires the guidance team to delineate the extracted causes, consequences, and feedbacks, and arrange them in CLD format (Fig. 5). The guidance team strives to ensure that all views are conserved and included in each individual CLD.

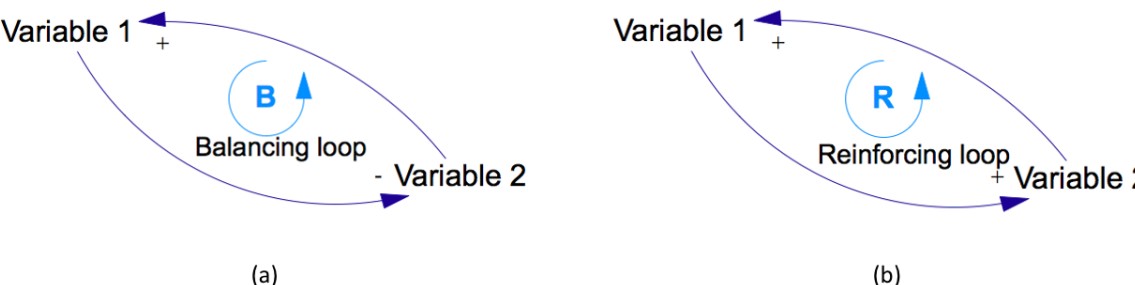

(a)                                                                                    (b)

**Figure 4: CLD: variables, causal links, and feedback loops**

Variable A often increases the Problem. Last year, Variable A was lower. I think it's because there wasn't much Variable B. Variable B usually decreases Variable A. Variable B is much lower near sites rich with Variable C and Variable D. The more Variable C and Variable D increase, the more Variable B decreases. After the Problem reached its peak, there was a great increase in Variable E. I think that Variable E is one of the major consequences of the Problem. As Variable E started increasing, I started seeing Variable F decrease. When Variable E reached its maximum, Variable F had been depleted. On the contrary, Variable C increased as a result of higher Variable E. In return a decrease in Variable F causes a decrease Variable D.

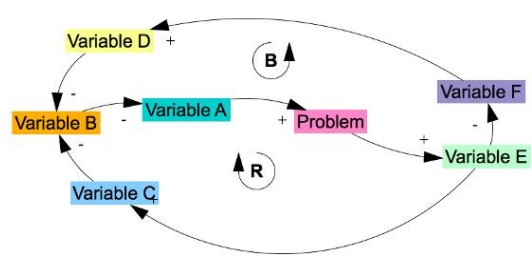

**Figure 5: A simplified version of a storyline and its corresponding CLD**

5.  Ensuring the conservation of all identified relationships, each individual CLD is joined, forming an overall merged CLD as per Inam et al. (2015).
6.  The merged CLD is (1) checked for inconsistencies or conflicts and (2) transformed into a storyline by listing the causes, consequences, and feedbacks contained by the CLD in a coherent and comprehensive narrative (Fig. 5).
7.  The modified storyline is translated into the languages considered in the model-building activity to make it more accessible to all stakeholders, including marginalized ones.
8.  A collective workshop or focus group discussion is held in which (1) the storyline is re-examined with stakeholders and compared with their expectations (Arico et al., 2001), and (2) associated inconsistencies and points of conflict (previously identified in Stage 2, step 5) are discussed with them. The storyline is then modified accordingly. The execution of multiple iterations between stakeholder consultations, storylines, and CLDs, as displayed in Fig.6, is recommended (Alcamo, 2008).

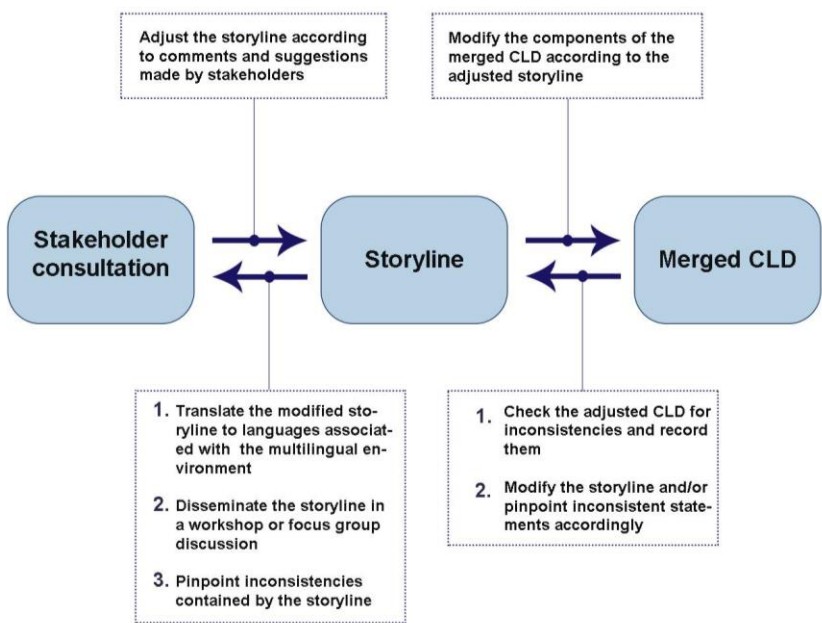

**Figure 6: Iterative process between stakeholder consultation, storyline development, and CLD construction**

9.  There are two outcomes to this stage: (1) a merged storyline to disseminate the results to marginalized stakeholders (specifically those with low literacy levels who might not be comfortable with the technicalities of CLDs) and (2) a merged CLD which is primarily used by the guidance team and associated researchers to visually identify feedback loops and facilitate the development of stocks and flows in later stages of the project.

10. The system's leverage points (e.g., balancing and reinforcing loops) and zones of undesired outcomes are identified after a merged storyline and corresponding CLD are agreed upon. It is important to note that this storyline and corresponding CLD represent the business-as-usual scenario, containing causes and consequences of the problem as-is without the implementation of policies or BMPs.

### 4.3  **Stage 3: Developing Micro-level Storylines**

1.  In a collective workshop, stakeholders are (1) addressed in the languages they speak and understand, and (2) grouped according to their preference towards receiving the results in CLD or spoken narrative form.

2.  Leverage points, such as balancing and reinforcing loops, and zones of undesired outcomes are outlined to stakeholders, highlighting targets for BMP and policy applications. Candidate policies that are capable of influencing highlighted targets (i.e., leverage points or undesired outcomes) are elicited from stakeholders.

3.  Members of the guidance team ask relevant questions to understand how the suggested policy or BMP either (1) reconfigures or restructures the system, or (2) weakens or reinforces aspects of it. The first part of each Micro-level storyline is comprised of the description of each suggested policy or BMP and how it can be integrated into the system.

4.  Participants are asked to describe how the implementation of suggested policies or BMPs changes the system's dominant state. In other words, they are asked to describe the future of the suggested policy or BMP in the context of the modelled problem. Elicited predictions, regarding each suggested policy or BMP, make up the second part of each corresponding Micro-level storyline.

5.  These policies and BMPs are then simulated in a quantitative version of the model. The results are subsequently presented to stakeholders by members of the guidance team, in the form of a comprehensive

narrative, to accommodate non-modelers and less literate stakeholders. These results are discussed until an agreement on suitable solutions is reached. This paper does not cover the implementation of this step.

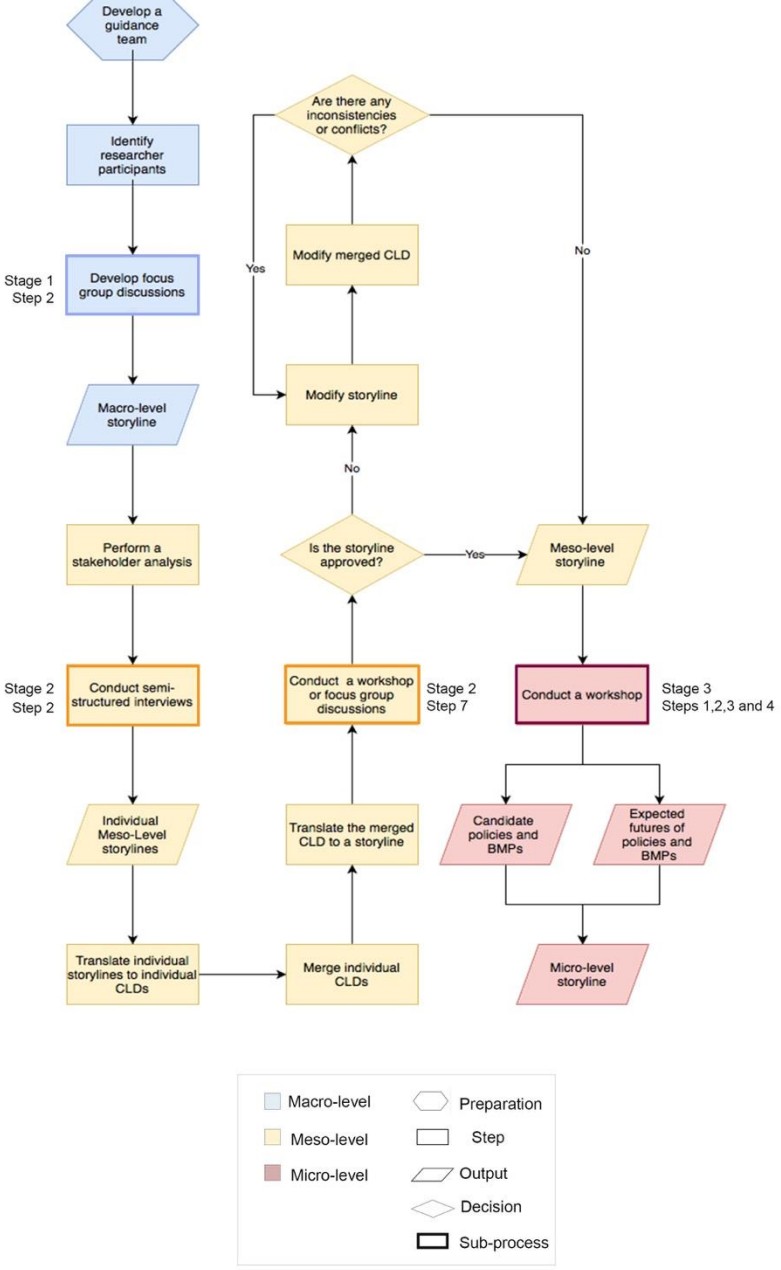

**Figure 7: Multi-level storyline development process**

## 5    Results

The Lake Atitlán case study examines the proposed framework's ability to engage stakeholders from the marginalized Mayan community in a participatory model-building activity to investigate the mechanisms governing cultural eutrophication in the area. Table 1 displays the demographics of the Atitlán watershed's general population

(INE,2018) and stakeholders who participated in the case study. A guidance team of three individuals with Kaqchikel, Tz'utujil, K'iche', and Spanish language skills was established before initiating the activity. All activities were carried out in relevant languages.

Members of the guidance team were aware that the activity presented a learning opportunity to them as well and remained cognizant of their positionality in the research setting. The priority of the guidance team was to create a space that allowed stakeholders to communicate their perspectives, needs, and concerns. This section provides an elaboration of extracted Macro-level, Meso-level, and Micro-level storylines. The authors highlighted three submodules (Fig. 8, Fig. 9, and Fig. 10), which are part of one conceptual model. The full model can be found in the supplementary material.

**Table 1: Demographics of project participants**

| Demographics | Participatory Modeling (%) | General Population of Tz'olöj Ya' (%) |
|---|---|---|
| **Women** | 24.1 | 52 |
| **Men** | 75.9 | 48 |
| **Indigenous** | 62.1 | 96 |
| **Kaqchikel** | 44.4 | 39 |
| **Tz'utujil** | 44.4 | 16 |
| **K'iche'** | 11.2 | 44 |
| **Hispanic** | 37.9 | 3 |
| **Indigenous language** | 58.6 | 81 |
| **Spanish language** | 41.4 | 18 |
| **Literate** | 86.2 | 70 |
| **Illiterate** | 13.8 | 30 |

## 5.1    Macro-level Storylines

The guidance team met with researchers from local and national, Indigenous and non-Indigenous, and academic and governmental institutions conducting research projects in the area. Researcher participants included individuals who identify as belonging to marginalized groups. Focus groups were held with researcher participants. When asked about an overarching problem in the Atitlán watershed, all parties mentioned the lake's eutrophication and associated water quality problems. The eutrophication of Lake Atitlán has been a pressing environmental problem for more than a decade. Researchers' interest in Lake Atitlán has increased since a major episode of cyanobacterial blooms covered 40% of the lake's surface in October 2009. This event impacted the activities in the area and received significant national and international media coverage. Moreover, the endorsement of the 'Mega-collector' by the government in 2018 reinforced the community's interest in the problem. All research participants have been working on projects associated with the lake's pollution.

Participants highlighted the dominance of three types of economic activities in the area: (1) agriculture, aquaculture, and tourism. They also delineated the presence of two types of authorities: Indigenous and non-Indigenous. For example, in Tz'olöj Ya' there are two municipalities, an Indigenous municipality and an official one. Nevertheless, the Indigenous municipality is not recognized by the Guatemalan government as the main authority but rather as auxiliary. In some towns, such as Pan Ajache'l and Tz'ikinajay, local Indigenous authorities called 'Cofradías' have power over local decision-making. However, a governmental institution remains the official authority for managing the Atitlán Basin. The area lacks a unified platform for decision-making, which restricts the

proper implementation of BMPs and policies. Therefore, different stakeholder groups have attempted to implement various remedies to improve the lake's water quality. However, their efforts had never been joined, failing to significantly impact the state of the lake. The contrasting perspectives of different stakeholders and the complex political culture of the area have been prominent barriers to the coordinated discussion and implementation of

sustainable solutions. Most researcher participants agreed that the eutrophication problem stems from the lack of unified attempts to restrict nutrient discharge into the lake. Furthermore, they emphasized that the success of bottom-up management strategies or policies that aim at controlling nutrient enrichment requires the collaboration of stakeholders with diverse views, backgrounds, roles, and capabilities, many of whom belong to Mayan communities.

The Atitlán watershed encompasses diverse communities with distinct cultural backgrounds. Non-

Indigenous stakeholders are primarily Spanish-speaking, and Indigenous Peoples have Kaqchikel, Tz'utujil, or K'iche' first languages. Many Indigenous persons do not communicate well in Spanish and are more comfortable using their native languages. However, Indigenous languages in the region often face discrimination. This is reflected in educational systems, where these languages are not usually acknowledged, even in areas where Indigenous communities are predominant (e.g., 96% of the population of the department of Tz'olöj Ya' is

Indigenous).

Researcher participants also highlighted some historical events that influenced the problem and associated reactions. For example, residents had first witnessed cyanobacterial blooms in the lake in 2008 and more extensive ones in 2009. These blooms increased residents' environmental awareness of the lake's unhealthy trophic state, triggering bottom-up stakeholder-led actions. Also, some stakeholders mentioned that two hurricanes, Agatha in

2005 and Stan in 2010, had caused damage to the lake's ecosystem. Finally, in 2017, the Mega-collector project (elaborated on in Sect. 3) was proposed to solve the lake's eutrophication problem, triggering tensions between various communities opposing or supporting the project.

Macro-level storylines showed how primary researcher participants chose to model the eutrophication problem of Lake Atitlán. The geographical scope of the model was limited to the Atitlán Basin, and stakeholders

from Indigenous and Hispanic origins were considered. Three major economic sectors and concomitant stakeholders were also considered for the model-building activity: agriculture, aquaculture, and tourism. Although Mayan communities make up the majority of the area, most of the past participatory activities in the basin have been in Spanish. From the background information given by participants on power imbalances in the area, and to address relevant power dynamics, the official languages of the model-building project (including internal communication

between the guidance team and researcher participants) were chosen to be Mayan languages. However, the Spanish language was still used to address the Hispanic community and include them in the process. Finally, the consideration of stakeholders from both official governmental institutions and local Indigenous authorities was deemed important.

**5.2     Meso-level Storylines**

The guidance team used information encompassed by Macro-level storylines about involved authorities, communities, and economic sectors in the area to identify relevant stakeholders. The initial list of stakeholders included Indigenous and non-Indigenous municipal authorities in the Atitlán watershed, local Indigenous authorities (i.e. Cofradías), relevant governmental institutions (the lake's authorities, environmental institutions, and

agricultural institutions), farmers' associations, fishers' associations, academic institutions, non-governmental organizations, community-based organizations, and owners of tourism businesses. To construct the Meso-level storylines, stakeholders were first informed of the problem and its background using the Macro-level storyline and then interviewed to elicit causes and consequences underpinning the problem (following the structure of a CLD construction process).

### 5.2.1 Causes

Members of the guidance team initiated each interview with the following focal question: What are the causes of the nutrient enrichment problem in Lake Atitlán? The majority of the interviewees listed soil erosion, inorganic agriculture, and untreated wastewater discharge as primary causes for nutrient enrichment. They attributed soil erosion to deforestation and the latter to urbanization, expansion of agricultural land, and forest fires. Most Indigenous participants stated that the lack of septic tanks and dry toilets exacerbated wastewater discharge. However, a mix of Indigenous and non-Indigenous participants attributed the latter to the lack of wastewater treatment (WWT) facilities, combined with an increase in population (Table 2). Many stakeholders also connected the dominance of inorganic agricultural practices in the area to the need for farmers to maximize profit and governmental subsidies on inorganic fertilizers, among other causes.

A few stakeholders cited inorganic soaps and detergents from people washing their laundry in the lake as a contributor to nutrient enrichment. Some participants linked the loss of native fish species, due to overfishing and invasive fish, to increases in nutrient concentration. Education and environmental awareness were also correlated to multiple variables each. For example, some stakeholders mentioned that an increase in the education level yields a decrease in population but an increase in environmental awareness. Subsequently, an increase in environmental awareness would lead to a decrease in the use of inorganic soaps and detergents. Moreover, participants connected different land-use variables (such as agricultural, forest, and urban areas) to nutrient concentration levels in Lake Atitlán (figures 8 and 9). For example, some stakeholders stated that an increase in population leads to an increase in urban areas, consequently yielding a decrease in available land per household for the installment of septic tanks or dry toilets. As mentioned earlier, this increases quantities of discharged wastewater and, consequently, nutrient concentrations in Lake Atitlán.

### 5.2.2 Consequences

In the second part of the semi-structured interviews, the guidance team used the single-driving force method to elicit the consequences of the nutrient enrichment problem. Participants were asked the following questions: (1) what happens if nutrient concentrations in the lake increase? (2) What happens if they decrease? All participants listed cyanobacterial blooms and the loss of biodiversity as direct consequences of nutrient enrichment of Lake Atitlán. Some stakeholders correlated cyanobacterial blooms to a decrease in tourism, resulting in less revenue for many businesses in the watershed. Other stakeholders mentioned that cyanobacteria would cause illnesses that would decrease workers' productivity, leading to the reduction of agricultural labor and cultivated areas. Others highlighted the effects of loss of fish species due to high concentrations of nutrients, consequently affecting the income of people involved in fishing. As mentioned by participants, the aforementioned indicates that an increase in nutrient enrichment leads to decreased economic prosperity in tourism, agriculture, and aquaculture. Some participants stated that high concentrations of nutrients render the lake's freshwater undrinkable, potentially leading to illnesses and loss of productivity in the area, in addition to increased use of plastic bottles.

### 5.2.3 Feedback Loops

The narration of consequences by stakeholders allowed for the identification of feedback effects. The most important feedback loops are contained by (1) two modules representing the local agriculture (Fig. 8) and tourism (Fig. 9) economic sectors and (2) one module representing the mechanisms governing environmental awareness in the region (Fig. 10).

Some feedback loops were described by stakeholders in terms of generalized relationships between nutrient enrichment and economic prosperity (Fig. 11). Feedback links between (1) farmer's income and education (B1, Fig. 8), (2) poverty and education (R5, Fig. 9), and (3) tourism business revenues and potential investments in wastewater treatment plants (WWTPs) (R6, Fig. 9) indicate that the relationship between nutrient enrichment in Lake Atitlán and economic prosperity is represented by a reinforcing feedback loop (Fig. 11 (a)). In other words, some stakeholders stated that economic prosperity (1) increases the education rate, which ultimately decreases

population and, subsequently, nutrient enrichment in Lake Atitlán, and (2) increases potential investments in WWTPs, reducing nutrient discharge into the lake. Those feedback effects were elicited from a mix of Indigenous and non-Indigenous stakeholders (Table 2).

On the contrary, relationships between (1) farmer's income and potential investments in improving irrigation efficiency (R1, Fig. 8), (2) farmer's income and potential investments in cultivated areas (R2, Fig. 8), and
(3) the number of tourists and the amount of discharged wastewater (B4 and B5, Fig. 9) portray feedbacks between nutrient enrichment in Lake Atitlán and economic prosperity in the form of a balancing loop (Fig. 11 (b)). Namely, some participants implied that economic activities generated by agriculture and tourism yielding economic prosperity (which is perceived by other stakeholders to provide the resources for education and technological investment for environmental improvement) are the primary causes of the nutrient enrichment problem. Since
economic prosperity reinforces economic activities (R1 and R2 in Fig. 8, R7 in Fig. 9), which are presently unsustainable, economic prosperity therefore exacerbates nutrient enrichment in Lake Atitlán. This balancing relationship between economic prosperity and nutrient enrichment was strictly obtained from the contribution of Indigenous participants (Table 2).

Conversely, stakeholders linked the dominance of cyanobacteria with environmental awareness. Balancing
loops representing this relationship (displayed in Fig. 10), were strictly elicited from members of civil society (NGOs and community-based organizations with Indigenous and non-Indigenous members) (Table 2).

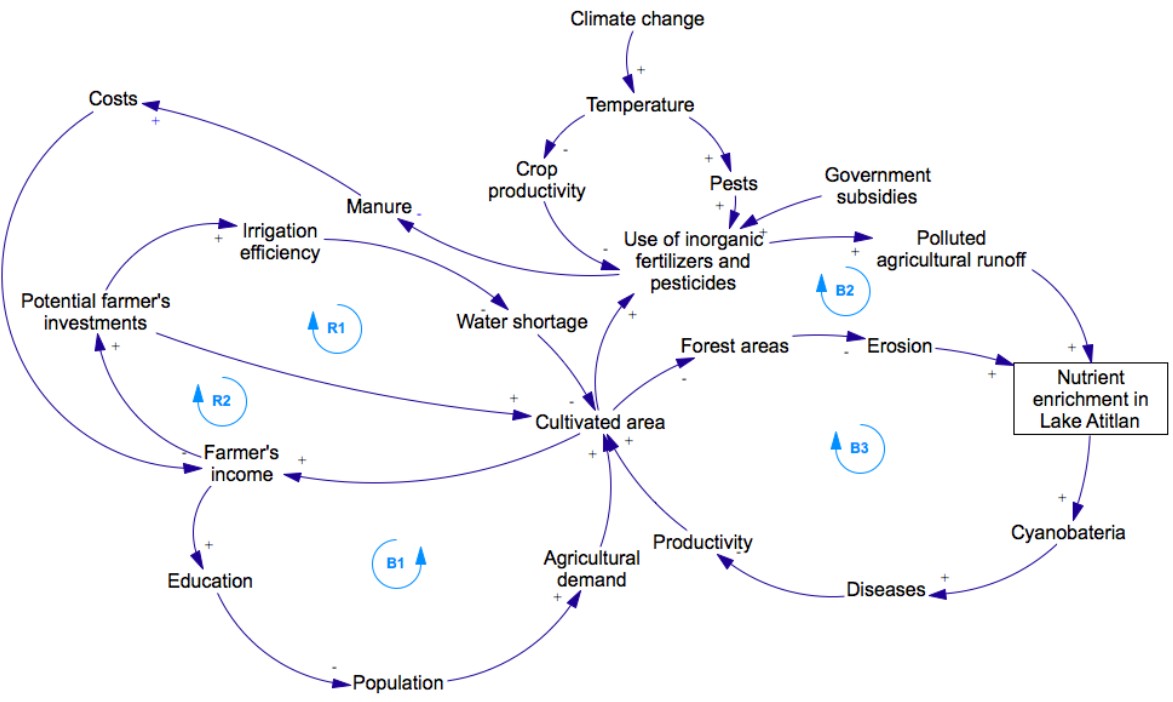

**Figure 8: Agriculture submodule**

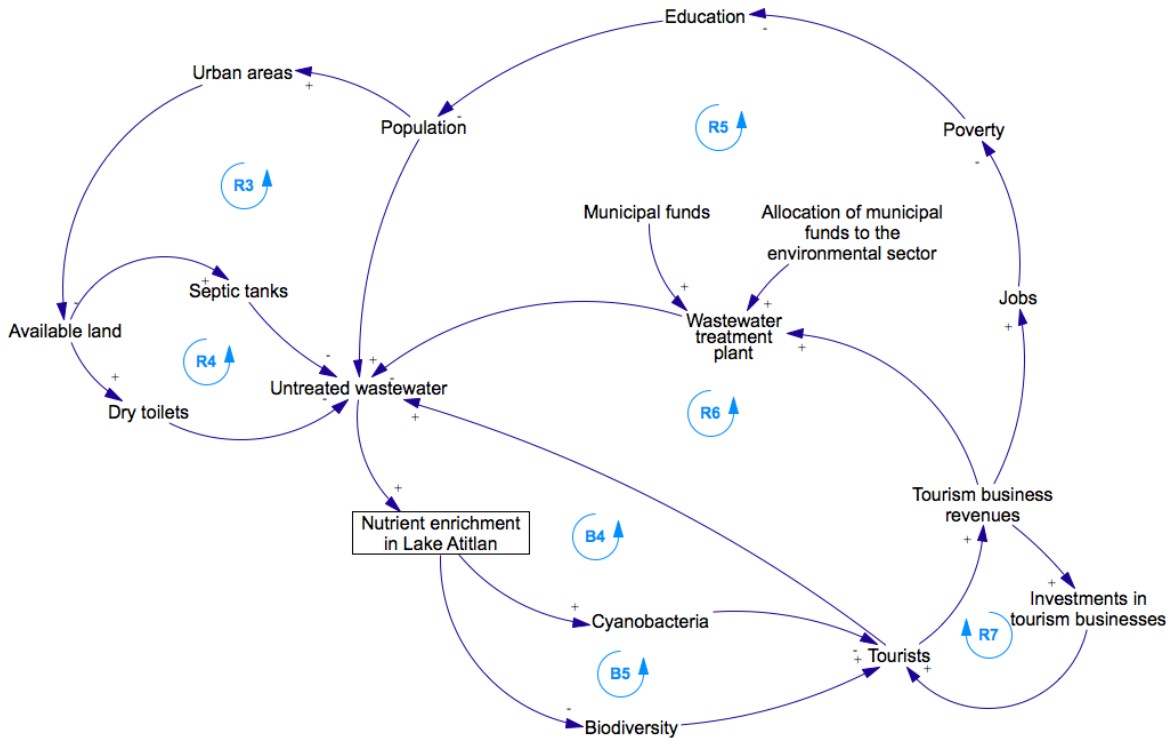

**Figure 9: Tourism submodule**

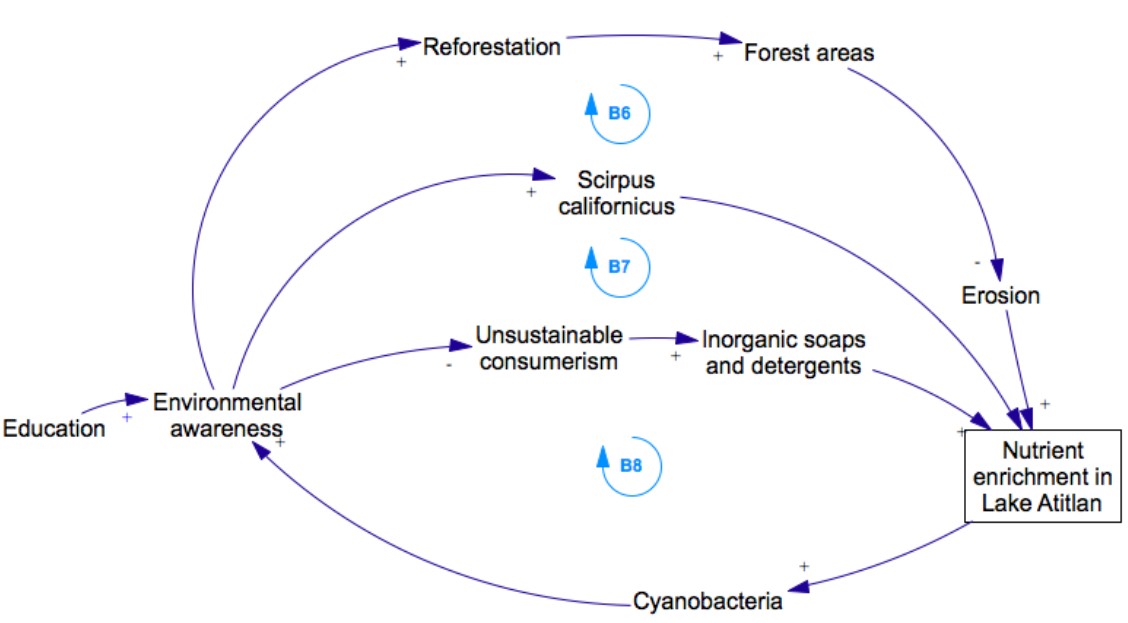


**Table 2: Highlights of unique contributions from diverse stakeholder groups**

| Contribution | Reference | Contributors |
|---|---|---|
| 'WWTP' variable | R6 in Fig. 9 | Mix of Indigenous and non-Indigenous participants |
| 'Dry latrines' and 'Septic tanks' variables | R3 and R4 in Fig. 9 | Indigenous participants |
| Feedbacks contributing to the reinforcing loop between nutrient enrichment in Lake Atitlán and economic prosperity (Fig. 11 (a)) | B1 in Fig. 8; R5 and R6 in Fig. 9 | Mix of Indigenous and non-Indigenous participants |
| Feedbacks contributing to the balancing loop between nutrient enrichment in Lake Atitlán and economic prosperity (Fig. 11 (b)) | R1 and R2 in Fig. 8, B4 and B5 in Fig. 9 | Indigenous participants |
| Balancing feedbacks between nutrient enrichment in Lake Atitlán and environmental awareness | B6, B7, and B8 in Fig. 10 | Civil society |
| Positive relationship between crop productivity and the use of inorganic fertilizers | Excluded (refer to Sect. 5.2.4) | Decision-makers |
| Negative relationship between crop productivity and the use of inorganic fertilizers | Fig. 8 | Agriculturists/farmers |

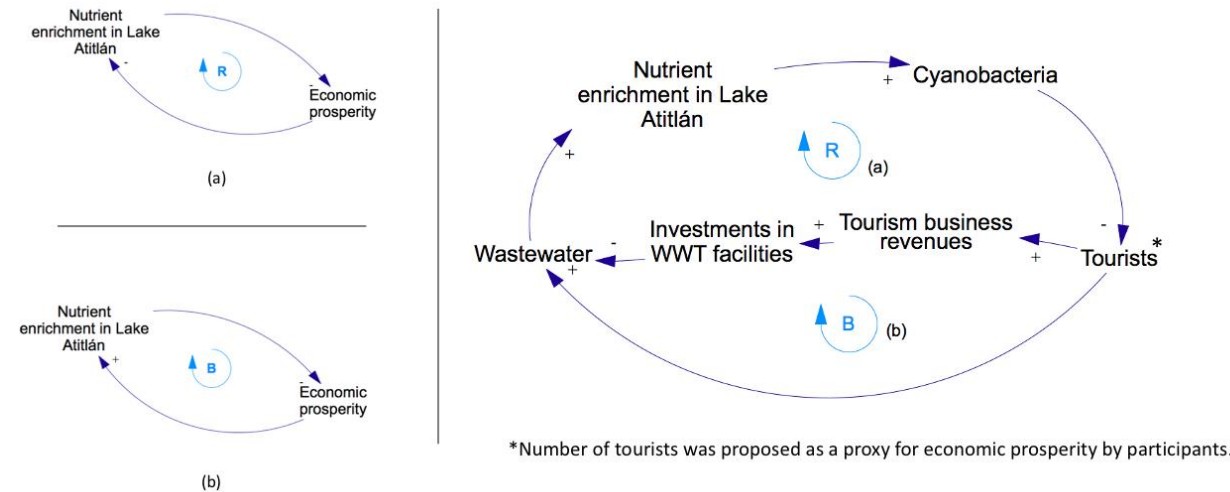

*Number of tourists was proposed as a proxy for economic prosperity by participants.

**Figure 11: The relationship between nutrient enrichment in Lake Atitlán and economic prosperity (reinforcing loop (a) and balancing loop (b)). The two loops on the left represent generalized relationships of the two loops on the right, mentioned and agreed upon by participants. The two contradicting views underpinning the two generalized relationships (loops (a) and (b) on the left) were elicited by different stakeholder groups. The delineation of both relationships shows that all potentially valid points can be represented explicitly in the model, which reinforces the point**

**of inclusivity. Quantification would show which of the two loops dominates the model's behavior.**

### 5.2.4    Points of conflict

Multiple points of conflict were detected and discussed with relevant participants (selected according to their relevance to the case-specific conflicts) to find solutions. For example, while farmers stated that a decrease in crop productivity and an increase in pests would drive farmers to use more inorganic fertilizers and pesticides, decision-makers suggested that they would make farmers shift to organic agricultural practices, seeking long-term benefits (Table 2). Members of the guidance team met with farmers to discuss this paradox and found that the actual barrier for the adoption of organic agricultural practices is economic. The majority of farmers in the area preferred rapid and short-term monetary benefits over the long-term advantages of organic agriculture. Therefore, as presented in Fig. 8, the relationship between crop productivity and the use of inorganic fertilizers is considered negative (Table 2).

Another point of misunderstanding was the relationship between income and investments in WWT facilities. Some participants stated that increased revenue from tourism, agriculture, and aquaculture leads to increases in potential investments in WWT facilities. Nevertheless, others emphasized the importance of distinguishing different sources of income and the relevance of these sources to the sectors responsible for investing in WWT facilities. They also highlighted that a significant barrier to the development and maintenance of WWT plants is the distribution of public funds. Regardless of the public sector's monetary capacity, an insufficient amount of funds is typically allocated to environmental management services, such as WWT facilities. After investigating these claims with employees in the tourism sector, an increase in tourism was considered to increase the tourism business owners' capacity to invest in on-site WWT systems. This has already been done in multiple hostels in towns around Lake Atitlán, such as Pan Ajache'l. Also, subnational governments (the official municipalities) are considered responsible for the construction of central WWTPs in towns contained by the watershed. Subnational governments corresponding to towns in the watershed receive the majority of their income from subsidies and grants. Therefore, an increase in subsidies and grants, coupled with increased allocation of funds to environmental services, is expected to increase the development of WWT facilities (Fig. 9).

### 5.3    Micro-level Storylines

A collective workshop was held to construct the Micro-level storylines. At first, the candidate solution, known as the Mega-collector project (discussed in Sect. 3), was the center of the policy discussion. There was a clear divide between stakeholders who supported or opposed the project. Stakeholders who advocated for the Mega-collector stated that wastewater discharge into the lake is the primary cause of eutrophication. Therefore, implementing the project would definitely decrease nutrient concentrations in Lake Atitlán. Those who were against the project stated that it (1) does not target major mechanisms contributing to the nutrient enrichment problem, such as agricultural runoff and erosion, and (2) eliminates dilution (which is essential for decreasing nutrient concentration) by diverting treated wastewater from the watershed. Moreover, one stakeholder highlighted that more than 60% of wastewater in the area is not discharged through a drainage system, meaning that the project would only target about 40% of produced wastewater. Some stakeholders, therefore, stated that the Mega-collector project would not be as effective in improving the lake's trophic state. Moreover, they emphasized that exporting water resources outside the watershed would exacerbate the water shortage problem. They also expected that the large-scale project would pose a threat to the lake's biodiversity (which is crucial to residents and businesses in the watershed). The opposition also cited public safety concerns since the area is bounded by seismic faults.

Different stakeholder groups suggested different policies and BMPs targeting various leverage points (e.g., reinforcing and balancing loops). Decision-makers reiterated the importance of developing WWTPs. While some suggested a centralized WWTP (resembling the Mega-collector project), others recommended a decentralized WWT system. Farmers focused on the importance of organic agriculture to reduce the discharge of polluted agricultural runoff into Lake Atitlán. They highlighted the importance of (1) economic incentives to align sustainable agricultural practices with farmers' goals of profit maximization and (2) good governance to align expected outcomes with actual results. They specified the significance of setting the variable 'Farmer's income' (Fig. 8) as an evaluation metric for relevant policies and BMPs, to ensure their cooperation. Fishers' associations suggested

imposing regulations for sustainable fishing practices and planting and preserving *Scirpus californicus*. They also emphasized that fishers' income should be an evaluation index for potential policies to ensure the collaboration of fishers and the aquaculture industry. Finally, members of the civil society highlighted the importance of forest preservation and reforestation initiatives to prevent eroded soils from entering the lake. When asked about the future of the policies and BMPs they recommended, stakeholders stated that they do not expect each policy or BMP to have a significant impact alone. However, they expect the collaboration between different sectors and the collective implementation of the mentioned policies and BMPs to decrease nutrient concentrations in Lake Atitlán.

## 6    Discussion

### 6.1    Evaluation

The purpose of this study was to show how integrating the multi-level storytelling technique into participatory model-building processes (1) facilitates the inclusion of marginalized stakeholders (less literate, relatively powerless, and associated with marginalized languages), (2) initiates a dialogue, (3) integrates different perspectives of the problem, (4) facilitates model conceptualization, and (5) yields a nuanced understanding of human-water feedbacks governing the investigated problem. The suggested methodology was able to incorporate participants of low literacy levels, which might not have been achieved using other methods. Participants who cannot read or write were able to convey information comfortably. Also, stakeholders were at ease during individual interviews, especially when the guidance team assured them of the confidentiality of their identities. This process succeeded in reducing unhealthy power dynamics and provided an opportunity for the participation of key stakeholders who usually exclude themselves from such activities due to power issues.

Moreover, the variety of relevant languages spoken by the guidance team and stakeholders' freedom to convey information in their preferred language allowed for the participation of numerous primary stakeholders whose first language was not Spanish (the language used in similar activities in the past). Additionally, Indigenous communities considered the use of Indigenous languages as official languages of the project to have greater implications (e.g. it increased their trust in the activity). Numerous Indigenous participants cited this as the primary reason for their participation. Indigenous communities had lost confidence in such processes, as they had witnessed the "tyrannical potential" of participatory activities (Cooke and Kothari, 2001) since previous participatory approaches in the area did not effectively incorporate them. Therefore, instead of effectively integrating Indigenous communities in decision-making, previously conducted participatory processes often reinforced illegitimate and unjust decisions, while claiming them as 'participatory.' The use of Indigenous languages by members of the guidance team and in documents, visual presentations, and workshops was key to gaining the trust of Indigenous communities. This trust triggered the willingness of some Indigenous participants to start a dialogue and communicate with other stakeholder groups. Carried out in a culturally relevant way, the participatory process allowed Indigenous communities and Hispanic stakeholders to discuss and share solutions during workshops.

The authors suggest that inclusiveness endorses equitable community-based decision-making. They also emphasize that fostering the inputs of marginalized stakeholders and inducing collaboration through inclusion is important for implementing successful solutions. This is evident by the significant contributions to the modeling process made exclusively by Indigenous participants. Exclusive contributions by different stakeholder groups, representing their unique perspectives, are displayed in Table 2. All these contributions were conserved and included in the conceptual model. Moreover, in many cases, similar to the demonstrated case study (e.g. Hassanzadeh et al., 2019; Izurieta et al., 2011), marginalized stakeholders are central to both the persistence and remediation of the examined environmental problem. Therefore, ensuring their inclusion in participatory model-building activities is crucial.

The construction of multi-level storylines also proved to be compatible with CLD development (which is important for conceptualizing systems models). Elicitation of the Macro-level storyline guided and informed the subsequent stages of the process and helped define the scope of the model and the variables and policy scenarios

within that scope. Meanwhile, the extraction of the Meso-level storylines helped develop an appropriate understanding of the relationships (causes, consequences, and feedbacks) governing the problem. Once the Meso-level was described, leverage points in the modelled system were explored by identifying critical balancing and reinforcing loops before considering BMPs and policy scenarios. Finally, the elicitation of Micro-level storylines aided in identifying potential BMPs and policies by targeting the leverage points and undesired outcomes mentioned above.

Quantification is needed to assess the impacts of suggested solutions. Nevertheless, some insights can be identified from the qualitative modeling exercise. For example, wastewater treatment, which was discussed by stakeholders, could play an important role in decreasing the discharge of untreated wastewater produced by residents and tourists (R6 in Fig.9). However, about 60% of wastewater in the area is not discharged through a drainage system (Romero, 2013). Therefore, contrary to what some stakeholders suggested, the proposed plan would not present an optimum solution unless coupled with other projects such as drainage system planning and dry toilets. On another note, aiming to reduce the consumption of inorganic fertilizers by supporting organic agriculture (as mentioned by participants in Sect. 5.3) could potentially decrease the contribution of agricultural activities to nutrient enrichment (Fig.8). In this light, subsidies on inorganic fertilizers present an interesting leverage point in the system. Reexamining subsidies and reallocating financial resources to incentivize organic agriculture might play a role in increasing the efficiency of fertilizer application and, consequently, decrease nutrient enrichment in the lake. Finally, the goal of the system is a potent leverage point (Fischer and Riechers, 2019; Meadows, 1999). In this case, rethinking the goal, which focuses on decreasing nutrient enrichment, might be useful. This was not explicitly mentioned by stakeholders as a solution but rather implicitly through discussions about the 'Mega-collector'. The 'Mega-collector' project was opposed by many stakeholders partially since they anticipate that, while addressing the lake's water quality problems, it could also lead to other problems (e.g., water shortage, economic disparities and loss of biodiversity). Therefore, shifting the goal of the system to focus on an environmental component that could offer a more holistic view of the system's wellbeing, such as biodiversity, might be useful.

On another note, this study has three main limitations. First, it is difficult to assess the inclusiveness of the process. For example, the authors considered unique contributions of different stakeholder groups to indicate inclusiveness; however, this might simply be an indicator of the complexity of the problem (Rowe and Frewer, 2004). Second, the process included individual sessions to reduce the  impact of unhealthy power dynamics and encourage the effective involvement of less-powerful participants (Inam et al., 2015). However, group sessions (e.g. workshops and focus groups) were needed to initiate a dialogue between different stakeholder groups (Evans, 2006). The guidance team tried to detect unhealthy power dynamics and designed the agendas of these group sessions to explicitly encourage the participation of less-powerful stakeholders. However, the extent to which unhealthy power relations impacted the effectiveness of participation was unknown. Finally, a feedback loop between crop productivity and use of inorganic fertilizers (Fig. 8) might exist. However, the mechanisms and nature of this loop have not been further explored due to time constraints.

## 6.2    Human-Water Feedbacks

Eliciting storylines from stakeholders helped detect human-water feedbacks, even more so than CLDs. When participants construct CLDs themselves, they are restricted by variables and causal links between them. Storylines allowed for narrating more nuanced versions of connections between variables. This prevented participants from making reductionist assumptions (typically resulting from the restrictive nature of CLDs) and allowed for relevant discussions. Dynamics of human-water feedbacks discussed by stakeholders were aligned with those mentioned in the literature: the Rebound Effect (Dumont et al., 2013) and the Pendulum Swing (Van Emmerik et al., 2014). This shows how storytelling is compatible with human-water systems; it facilitated the capture of abstract concepts encompassed by human-water feedbacks that might not have been identified using other model-building methods or data sources. The identification of relationships that have been observed or pointed out by previous studies is valuable to the advancement of the study of human-water systems.

The Rebound Effect describes the appearance of unintended outcomes resulting from the implementation of
technocratic solutions that fail to consider sociocultural factors (Di Baldassarre et al., 2019). More specifically, it
states that the application of technologies to increase efficiency in resource use often increases resource
consumption (Alcott, 2005; York and McGee, 2016). An example of the Rebound Effect, known as the irrigation
paradox (Dumont et al., 2013), was highlighted by stakeholders. Numerous participants questioned the assumption
that an increase in farmers' technological investments in irrigation efficiency would definitively reduce agricultural
runoff. While water shortage is a dominant problem in the region's agricultural sector, most participants agreed that
increased irrigation efficiency would lead to the expansion of cultivated land. The saved water would thus be
reallocated by farmers to cultivate more crops and irrigate larger areas (Fig. 8). The latter has been confirmed by
earlier discussions with farmers, who claimed to favor profit maximization. The information elicited by the proposed
methodology allowed for the consideration of expected farmers' behaviors and navigation of commonly made
assumptions that contradict them. This is important for robust decision-making in water resources management,
since ignoring behaviors when creating solutions can lead to unintended socioeconomic feedbacks that lessen or
reverse the intended impact. In other words, acknowledging relevant sociocultural behaviors using unconventional
methods, such as storytelling, might help ensure that the actual outcomes of corresponding solutions are consistent
with predicted ones.
The Pendulum Swing (Van Emmerik et al., 2014; Liu et al., 2015) is described as the change of priorities
from immediate economic prosperity to environmental protection or vice versa (Di Baldassarre et al., 2019). This
phenomenon was delineated by several stakeholders and represented in two different balancing loops (B6 and B7 in
Fig. 10). Central to the representation of this phenomenon was the concept of environmental awareness, which was
mentioned by many stakeholders in this study and highlighted in previous models (e.g. Van Emmerik et al., 2014).
For example, stakeholders stated that the major cyanobacterial blooms in 2009 increased environmental awareness
in the area. Prior to the blooms, practices encouraged the expansion of agricultural areas through deforestation.
However, after the symptoms of the lake's degradation appeared, extensive reforestation campaigns were initiated
by the government to prevent soil erosion. Therefore, the cyanobacterial blooms caused a shift to prioritizing forest
over agricultural areas. The cyanobacteria bloom also spurred fisher-led campaigns for the restoration and protection
of *Scirpus californicus* along the lake's borders, which had been overexploited for craft production and destroyed by
hurricanes Stan (2005) and Agatha (2010). Through these examples, it can be seen that storylines can complement
datasets and quantification processes. Elicited explanations, such as expected changes in forest areas, could enable
robust projections of data trends, explain fluctuations in data trends, and facilitate the conceptualization and
projection of relationships contained by the model.
The generated model also reflects a more general conflict over the relationship between environmental
degradation and economic growth. Mechanisms that create reinforcing feedbacks (e.g., R6 in Fig. 9) and balancing
feedbacks (e.g., B5 in Fig. 9) between factors indicative of economic growth (e.g., revenue and investments) and the
lake's trophic state were elicited from stakeholders. As mentioned earlier, for example, while some stakeholders
suggested that tourism activities yielded mechanisms exacerbating the lake's trophic state, others highlighted the
need for revenues generated by such activities to invest in technological facilities to improve the lake's water quality
(i.e., WWTPs). This indicates that the applied method was capable of organically capturing the archetypal debate,
surrounding the relationship between environemtnal degradation and economic growth, through diverse
socioculturally explicit perspectives. This is crucial for (1) modeling human-water systems, where different
governing sociocultural mechanisms require more nuanced versions of generalized relationships and (2) developing
well-targeted recommendations in water resources management. For example, in this case study, including a
contextualized version of the relationship between economic prosperity and nutrient enrichment of the lake allows
the development of relevant recommendations that aim to (1) intensify the impact of the reinforcing loop (e.g.
optimize the allocation of resources generated by economic prosperity to reduce nutrient enrichment in the lake) and
(2) abate the impact of the balancing loop (e.g. ensure that economic prosperity is driven by environmentally
sustainable economic practices that have no or minimal adverse effects on Lake Atitlán) by targeting the
socioculturally specific mechanisms that govern each.

## 7 Conclusion

The proposed participatory model-building framework helps to address the challenges of tailoring PM activities in water resources management to accommodate diversity within societies and facilitate the inclusion of marginalized stakeholders (i.e., less literate, comparatively powerless, or associated with marginalized languages). In general, many PM processes' implementation remains biased as they often view communities as homogeneous units and do not consider different capabilities, needs, and interests within diverse communities.

The authors suggest that storyline development is capable of facilitating inclusiveness in participatory modeling. However, since the literature on PM in environmental and resource management contexts primarily provides participatory storyline development methodologies that (1) are either compatible with the development of linear models, or (2) do not expose the leverage points of the system prior to selecting and testing relevant solutions, the authors propose a conceptual framework for developing storylines that aim to conceptualize and inform systems models while making use of the systems' leverage points. The proposed framework is underpinned by the MLP framework, adjusted to accommodate the conceptualization of multi-level storylines. The authors then offer a stepwise approach for implementing the process while helping to facilitate the inclusion of marginalized stakeholders.

The proposed framework was tested in the Atitlán Basin, Guatemala and aimed to incorporate marginalized Mayan communities in the PM process. The applied method was able to (1) incorporate stakeholders who are less literate, relatively powerless, and associated with a marginalized language in the PM process, and (2) integrate different perspectives of diverse community members. Results showed that not only is inclusiveness important to endorse equitable decision-making, but it also (1) fosters key inputs from marginalized stakeholders and (2) induces the needed dialogue for the successful implementation of solutions. Moreover, the method provided stakeholders with an opportunity for narrating more nuanced versions of relationships between variables, allowing the extraction of contextualized human-water feedbacks.

The suggested conceptual framework facilitated the translation of storylines into relationships that form the conceptual basis of the systems model. As a next step, the conceptual model can be transformed into stocks and flows and quantified. The quantified model would be inherently underpinned by socioculturally specific relationships and, therefore, could help decision-makers develop well-targeted recommendations in water resources management.

## 8 Author Contributions

JBN developed the methods, carried out field work, performed the analysis, and wrote the paper. JM and MRR conducted field work. JA, JM, HT, and WM supervised the research. The authors declare no conflicts of interest.

## 9 Acknowledgements

This paper draws on research supported by the Social Sciences and Humanities Research Council of Canada (SSHRC 435-2017-1482). The authors thank Ms. Emma Anderson for editing the paper and providing helpful comments. This research was approved by the Research Ethics Board Office in the Faculty of Agricultural and Environmental Sciences, McGill University.

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
