# Peer review of "Multi-level storylines for participatory modeling — involving marginalized communities in Tz'olöj Ya', Mayan Guatemala"

_Hydrology and Earth System Sciences, 2020_

## Referee Comment (RC1) · Anonymous Referee #1 · 27 Sep 2020

**Multi-level storylines for participatory sociohydrological modelling – involving marginalized communities in Tz'olöj Ya', Mayan Guatemala**

**Jessica A. Bou Nassar et al.**

**Scientific significance**

The manuscript represents a contribution regarding the implementation of participatory modelling in vulnerable and disadvantaged communities to address eutrophication problems. Understanding and applying tools such as participatory system dynamics in different contexts is illuminating for research and practice and therefore, the manuscript offers a valuable work to be published in this special issue of the journal.

The authors present the integration of multi-level storylines in participatory modelling as the main contribution and as novel feature that departs from traditional participatory modelling approaches. However, the methodology and results associated to the multilevel storylines is really similar compared to the initial phases of conducting traditional participatory modelling processes to elicit causal loop diagrams. In this regard, it is required to strength the comparison of traditional approaches to elicit causal loop diagrams in participatory process and the storylines approach, or otherwise, presenting the storylines in their fair dimensions as an alternative to elicit causal loop diagrams in participatory process.

**Scientific quality**

The authors reviewed relevant literature regarding the building blocks of the approach adopted. Nonetheless, participatory modelling has been used in water resources management almost from its beginning and some relevant authors regarding the integration of these concepts were overlooked what perhaps lead the authors to not sufficiently acknowledge that practices quite similar to those developed in their research have been employed to: i) construct causal diagrams eliciting the stakeholders´ perspectives; and ii) take part in the whole cycle of system dynamic modeling.

**Presentation quality**

The manuscript structure could be substantially improved. There are different ways the manuscript could be better structured. One could be to show a conceptual framework with the original building blocks, followed by a proposal of the integrated approach. Then, the methodology where this integrated approach is materialized could be described. After this, the results could be presented. The current way in which the document is developed, with parts of the original approaches and the integrated approach appear in the introduction, the background and then the methodology, is not clear. Another path could be to develop the three proposed objectives in the results section that currently only addresses the implementation of integrated approach. In any case, sections 4.1 and 4.2 should not be part of the methodology since those are

general descriptions of the approaches and not an account of the activities carried out to undertake the research process.

In addition, more insight should be provided regarding the case selection, and how the research idea and problem emerged from the interaction between researchers and stakeholders.

**Language**

I cannot comment on the language as English is not my mother tongue.

**Specific comments**

Lines 145 -150

*Another approach, stakeholder created causal loop diagrams (CLDs), contain variables connected by links indicating causal relationships. Although CLDs have been previously applied in participatory research (Inam et al., 2015, 2017b), their construction requires reading and writing skills. Hence, they are ill-suited for involving less-literate participants in participatory model-building activities.*

This statement is not necessarily true. It is not necessarily the stakeholders those who formulate the CLDs. This approach of a facilitator building CLDs from interviews or focus groups and "translating" the information provided into the System Dynamics language is widely used in SD in WRM.

Line 269

*Figure 1 Location of the study area in Guatemala. Created in QGIS (https://qgis.org/) using Esri (2009).*

The map in Figure 1 needs to be substantially improved. The location of the study area in Guatemala and of Guatemala in America must be shown. Labels relevant only to the study area with Font of an appropriate size together with a grid of coordinates should be included.

Lines 401 - 403

*The sign corresponding to each link indicates the type of relationship between the two variables: (+) indicates a direct relationship, while (-) implies an indirect one.*

This statement must be reworded. That is not the correct explanation of polarities for the causal relations in System Dynamics.

Lines 452 - 455

*These policies and BMPs are then simulated in a quantitative version of the model. The results are subsequently presented to stakeholders by members of the guidance team and discussed until an agreement on suitable solutions is reached. This paper does not cover the implementation of this step.*

Since a relevant feature of this work is the context of their application, involving marginalized and indigenous communities, it would be an important contribution to explain how model results were discussed and communicated to these stakeholders, which is relevant since the authors expressed that most of them cannot read or write.

Line 474

*Table 1: Demographics of project participants*

It would be convenient to expand the information on the number of participating indigenous communities and their different languages

Lines 511 – 512

*From the Macro-level storylines elicited from primary researcher participants, the authors concluded that the model should address the eutrophication problem of Lake Atitlán*.

According to that statement ¿how this research fits within a real participatory approach in which external agents (researchers) should have a facilitating role in a process through which the relevant stakeholders reflect, deliberate and are empowered to make decisions, rather than a role of extracting information from stakeholders and make decisions for them?

Figure 7

Assess whether there is a feedback loop between "crop productivity" and "use of inorganic fertilizers and pesticides"

Improve the figure so that the polarity between "irrigation efficiency" and "untreated wastewater can be observed".

Figure 8

Improve the figure so that the polarity between "WWTP" and "untreated wastewater" can be observed.

Use the term in full for WWTP

Improve the figure so that the polarity between "available land" and "septic tanks" can be observed.

correct typo in septic tanks

Correct the polarity between Jobs and Poverty. This polarity should be negative, not positive

Figure 11

Is confusing that a relation can be reinforcing and balancing at the same time. Please clarify.

---

## Referee Comment (RC2) · Anonymous Referee #2 · 12 Oct 2020

The authors present a method for participatory modelling for system dynamics models. The topic is interesting and potentially a nice contribution to the existing literature, but I do think the manuscript needs significant improvements. The authors provide a nice overview of participatory approaches and the limitations of current methods, however, to me it does not become entirely clear how the method applied in this research is different from already existing methods. It seems like the main difference is the fact that the authors used indigenous languages for conducting their interviews, which does not really make it a new method. According to the authors, the new framework should be able to "(1) incorporate effective participation of marginalized stakeholders, (2) induce collaboration, (3) integrate diverse perspectives, (4) facilitate model conceptualization

and (5) produce description of relevant socio-hydrological phenomena." Point 1 is addressed partly by using the native language of participants but I would expect that this would also be addressed in the stakeholder selection process, for example, how do you ensure that these marginalized stakeholders are not left out? The process described in section 4.3 states that in stage 1 one starts with developing a focus group with primary stakeholders, how did the researchers make sure these stakeholders were representative? Point 2 is not really addressed in the rest of the manuscript. How do the authors ensure that the proposed framework induced collaboration? Did this work? Did collaboration increase after the participatory modelling exercise? Point 3 is discussed a bit more, in the sense that the storylines of different stakeholders allow for different perspectives, but it is not clear how the different perspectives are integrated into one conceptual model and how in this process it is ensured that the views of marginalized stakeholders do not get lost. Point 5 is only discussed at the end of the manuscript in the discussion and it is not clear from the start what socio-hydrological phenomena are and why it is important that the participatory modelling process produces descriptions of phenomena.

The results section is very unclear to me. First of all the authors should perhaps check the system dynamics literature again for a clear description of a causal loop diagram. The authors mention that a plus indicates a direct relationship and a minus an indirect relationships. In system dynamics a plus usually denotes a positive causal relationship (i.e. if the influencing variable increases the influenced variable also increases) and a minus a negative causal relationship (i.e. if the influencing variable increases the influenced variable decreases). Also the authors' description of feedback loops is a bit confusing. In system dynamics the feedback loops are the loops that are indicated with B1, R1, etc. However, the authors seem to reference another feedback loop that consists of multiple feedback loops, it is not very clear what this means. In general it is not clear to me how the causal loop diagrams in Figure 8, 9 and 10 are related to each other. Are they submodels of the main conceptual model? In that case, it would be good to explain how the different submodels are connected. Or are they three different

models that each provide a potential explanation for the model, based on different storylines? Also, the results section does not describe very clearly what the different storylines are that came out of the participatory process and how they were integrated and translated into these conceptual models. The function of Figure 11 is not clear to me. Is this a simplified version of the conceptual model? Did all the stakeholders agree to this simplified version?

In the discussion the authors discuss two socio-hydrological phenomena that are relevant for the case study. This is a bit disconnected from the rest of the study. Why is this relevant? And if it is relevant, this should be discussed earlier on in the paper. I expect the discussion to be focused on how the proposed framework and the implementation of this case study succeeded (or not) in addressing the limitations of other approaches of participatory modelling, how it is able to address the above mentioned five points and what the limitations are of the framework and methods proposed in this study.

More general, the authors refer to the use of participatory approaches for system dynamics modelling, to me system dynamics modelling suggests the actual translation of the conceptual model in a quantitative version and running the model to check if the outcomes are correct and what is expected. The authors state that the final step of stage 3 is to simulate the policies with the model and discuss this with the stakeholders, however, in my opinion, translating the model into a quantitative version and running simulations should already be done in stage 2, as a check, to make sure the conceptual model makes sense, and model simulations could also help the stakeholder discussions about whether the model accurately represents the situation. I would suggest to include the quantitative model and relevant simulations in the manuscript. If not, I would suggest to rephrase the manuscript to use conceptual models instead of system dynamics models. Since, I think the conceptual loop diagrams that are developed in this paper are conceptual models of the reality but not yet system dynamics models.

[Figure]

437, 2020.

---

## Author Comment (AC1) · 17 Nov 2020

**Manuscript hess-2020-437**
**Response to Referee #2**

Dear Referee #2:

We are grateful for your valuable comments. Your feedback will help us improve our manuscript significantly. Please find below our responses to your comments.

**Acronyms**
**RC** - Referee comments
**AR** - Author responses

**Comments and responses**

1. **RC**: The authors present a method for participatory modelling for system dynamics models. The topic is interesting and potentially a nice contribution to the existing literature, but I do think the manuscript needs significant improvements.

   **AR**: Thank you! We appreciate your constructive feedback that provides valuable improvements to our manuscript.

2. **RC**: The authors provide a nice overview of participatory approaches and the limitations of current methods, however, to me it does not become entirely clear how the method applied in this research is different from already existing methods. It seems like the main difference is the fact that the authors used indigenous languages for conducting their interviews, which does not really make it a new method.

   **AR**: The approach we suggest is useful as it builds upon CLD construction methods to include more stakeholders meaningfully. We do not perceive it as an entirely new framework but rather as an extension to CLD building that can be implemented within marginalized communities. As pointed in the methodology, our research included iterations between storylines and CLDs. Storylines were used for two purposes:

   - **Extraction of information:** by definition, a storyline describes cause-and-effect relationships between events that impact certain components or actors. Therefore, storylines are compatible with CLDs. The main difference is that storylines provide more leeway for stakeholders to explain their inputs. For example, some stakeholders used metaphors or anecdotes to describe their observations. This is useful in the contexts of (1) less-literate and non-expert stakeholders who (a) might not be able to explicitly place their observations in the context of variables and links and (b) might feel intimidated by the technicalities of the CLD approach, and (2) Indigenous stakeholders who consider storytelling as a way to share knowledge. Although labelled as a 'simplified version of a storyline', we think that Figure 5 might be oversimplifying and misconstruing the

flexibility of storylines. Therefore, we will improve the figure to include the intricacies of an extracted storyline.

- **Dissemination of results and science communication:** disseminating results in the form of storylines is more suitable for an audience of non-experts especially in the context of marginalized communities that include stakeholders who might not be comfortable with deciphering CLDs.

In the context of results, the difference lies in the ability of the methodology to (1) accommodate marginalized stakeholders who might have not been able to effectively participate otherwise and (2) the unique contributions of those stakeholders.

To address your comment, we will highlight the above mentioned points, eliminate terms and phrases that might exaggerate the novelty of the method (e.g. new framework), and emphasize that  storylines used in parallel with CLDs allow for more inclusive stakeholder participation.

3. **RC**: According to the authors, the new framework should be able to "(1) incorporate effective participation of marginalized stakeholders, (2) induce collaboration, (3) integrate diverse perspectives, (4) facilitate model conceptualization and (5) produce description of relevant socio-hydrological phenomena."

   **AR**: Our responses to each of these points are found below: points 4 to 7.

4. **RC**: Point 1 is addressed partly by using the native language of participants but I would expect that this would also be addressed in the stakeholder selection process, for example, how do you ensure that these marginalized stakeholders are not left out? The process described in section 4.3 states that in stage 1 one starts with developing a focus group with primary stakeholders, how did the researchers make sure these stakeholders were representative?

   **AR**: Thank you for your comment. First, primary stakeholders (or researcher participants) included Indigenous stakeholders. Second, the guidance team (made up of three individuals in total) also included an Indigenous researcher from Tz'oloj Ya', associated with Universidad Rafael Landívar. Third, during the focus group discussion, the guidance team explicitly addressed the socio-cultural dimensions of the Lake Atitlan Basin and included stakeholders that represent those dimensions. Fourth, the guidance team was actively seeking Indigenous stakeholders and institutions (including traditional councils and youth groups). We will emphasize and elaborate the aforementioned points in the text.

5. **RC**: Point 2 is not really addressed in the rest of the manuscript. How do the authors ensure that the proposed framework induced collaboration? Did this work? Did collaboration increase after the participatory modelling exercise?

**AR**: Thank you! We agree that we will need to change the wording from 'induced collaboration' to 'induced a dialogue'. The participatory activity allowed different stakeholder groups, Indigenous and Hispanic stakeholders to discuss, propose, and share solutions in the two workshops. Establishing a sense of trust with the Indigenous community and gaining their confidence required the implementation of a process that was truly tailored to those communities. Multiple stakeholders stated that they had lost confidence in such processes since previous participatory approaches in the area did not effectively incorporate them. As stated in Lines 685-690: "Instead of effectively integrating Indigenous communities in decision-making, previously conducted participatory processes often reinforced illegitimate and unjust decisions, while claiming them as 'participatory.'" Hence, the willingness of some indigenous stakeholders to start a dialogue and communicate with other stakeholder groups towards finding solutions was triggered by the process which is inclusive by design and was conducted in a culturally relevant way. We will highlight the aforementioned in the text.

6. **RC**: Point 3 is discussed a bit more, in the sense that the storylines of different stakeholders allow for different perspectives, but it is not clear how the different perspectives are integrated into one conceptual model and how in this process it is ensured that the views of marginalized stakeholders do not get lost.

   **AR**: The merged storyline and CLD contains all variables and relationships extracted from stakeholders. As shown in Table 2 (Line 605), unique contributions of different stakeholder groups were pointed out, discussed, and included in the model. To emphasize this point, we will highlight the contributions of marginalized stakeholders and elaborate the table in the text.

7. **RC**: Point 5 is only discussed at the end of the manuscript in the discussion and it is not clear from the start what socio-hydrological phenomena are and why it is important that the participatory modelling process produces descriptions of phenomena.

   **AR**: First, we would like to point out that we will eliminate the term 'socio-hydrology' since it can be interpreted differently by different researchers which might cause confusion. What we are trying to address is the broad space of human-water interactions and therefore, to make the manuscript clearer, we will replace the term 'socio-hydrology' with 'human-water systems'. Second, we perceive the discussion of results in the context of human-water systems to be important since: (1)Dynamics of environmental awareness and the rebound effect have been extensively discussed by stakeholders and could be better explained using the concepts of human-water relationships in the existing literature. (2) Stakeholders' conceptualization of human-water relationships were aligned with those mentioned in the literature. Delineating relationships that are pointed out by observations from past studies is valuable to the advancement of the study of such systems. We will emphasize the importance of human-water relationships in the introduction and in the discussion.

8. **RC**: The results section is very unclear to me. First of all the authors should perhaps check the system dynamics literature again for a clear description of a causal loop diagram. The authors mention that a plus indicates a direct relationship and a minus an indirect relationships. In system dynamics a plus usually denotes a positive causal relationship (i.e. if the influencing variable increases the influenced variable also increases) and a minus a negative causal relationship (i.e. if the influencing variable increases the influenced variable decreases).

   **AR**: Thank you for pointing this out. We will reword accordingly.

9. **RC**: Also the authors' description of feedback loops is a bit confusing. In system dynamics the feedback loops are the loops that are indicated with B1, R1, etc. However, the authors seem to reference another feedback loop that consists of multiple feedback loops, it is not very clear what this means.

   **AR**: We will change Figure 4 to include 1 balancing loop and 1 reinforcing loop and refer to them in the text after describing balancing loops and reinforcing loops, respectively.

10. **RC**: In general it is not clear to me how the causal loop diagrams in Figure 8, 9 and 10 are related to each other. Are they submodels of the main conceptual model? In that case, it would be good to explain how the different submodels are connected. Or are they three different models that each provide a potential explanation for the model, based on different storylines?

    **AR**: They are submodules of one conceptual model. We will provide a link to the complete conceptual model.

11. **RC**: Also, the results section does not describe very clearly what the different storylines are that came out of the participatory process and how they were integrated and translated into these conceptual models.

    **AR**: Thank you for your comment. To address your comment, first, in the appendix, we will provide a few individual storylines. Second, the results section summarizes findings from the storylines. We will emphasize that in the text. As pointed out in the methodology section:

    (1) Macro-level storylines set the context of the conceptual model by (a) informing the stakeholder analysis and (b) providing interviewed stakeholders with background information that contextualize Meso-level storylines.

(2) Informed and contextualized by the Macro-level storyline, Meso-level storylines were extracted from stakeholders. Afterwards, each storyline was translated to a CLD. Individual CLDs were merged (forming a merged CLD) and then translated to a merged storyline for the purpose of dissemination - i.e. communicating the results with marginalized stakeholders (figures 5 and 7). The process was iterative until consensus on the merged storyline was reached (Figure 6).

(3) The sub-modules in figures 8,9, and 10 are the result of steps (1) and (2).

(4) Micro-level storylines were extracted to provide potential solutions to weak zones exposed by the conceptual model (communicated to less-literate stakeholders using storytelling). They were not incorporated in the current conceptual model since scenario simulation is outside the scope of the study.

12. **RC**: The function of Figure 11 is not clear to me. Is this a simplified version of the conceptual model? Did all the stakeholders agree to this simplified version?

    **AR**: Thank you for your comment. Figure 11 is not a simplified version of the conceptual model. It displays a **generalized relationship** between economic prosperity and nutrient enrichment  and is used to highlight feedback loops. The CLD on the right shows 2 loops: 1 balancing and 1 reinforcing.

    For both loops, and as explained in the Consequences section (Lines 555-565), the causal link corresponding to the impact of nutrient enrichment on economic prosperity is negative. This causal link is generalized and does not contain intermediaries since the point of the figure is to elaborate on the feedback (i.e. the impact of economic prosperity on nutrient enrichment).

    Figure 11 (a): Some stakeholders stated that economic prosperity increases potential investments in WWTPs which reduces the discharge of untreated wastewater, consequently decreasing nutrient enrichment. This decrease in nutrient enrichment would lead to an increase in economic prosperity. The causal link corresponding to the impact of economic prosperity on nutrient enrichment is negative. Therefore, the relationship between economic prosperity and nutrient enrichment in this case is represented by a reinforcing loop (Fig. 11 (a)).

    Other participants implied that economic prosperity increases investments in tourism businesses, which increases the number of tourists, consequently increasing the amount of untreated wastewater. This leads to an increase in nutrient enrichment which would cause a decrease in economic prosperity. The causal link corresponding to the impact of economic prosperity on nutrient enrichment is positive. Therefore, the relationship between economic prosperity and nutrient enrichment in this case is represented by a balancing loop (Fig. 11 (b)).

Both processes were the result of the inclusive participatory process and show the added value of incorporating marginalized stakeholders since the balancing loop between the two variables was exclusively identified by Indigenous stakeholders. Additionally, the delineation of both relationships shows that all potentially valid points can be represented explicitly in the model (which reinforces the point of inclusivity). However, we acknowledge that one of the two loops will dominante model behaviour. This will depend on model quantification.

To make the figure clearer and less confusing we will:
- Add intermediaries to the causal link corresponding to the impact of economic prosperity on nutrient enrichment
- Provide a clearer explanation in the caption
- Replace the current example (the loops on the right) with two examples mentioned within lines 570-587 and refer to the figure right next to the examples it represents
- Emphasize that the figure represents a generalized relationship
- Mention that model quantification will show which of the two loops will dominate model behaviour

13. **RC**: In the discussion the authors discuss two socio-hydrological phenomena that are relevant for the case study. This is a bit disconnected from the rest of the study. Why is this relevant? And if it is relevant, this should be discussed earlier on in the paper.

    **AR**: To respond to this comment, we reiterate our response to point 7 above:
    "First, we would like to point out that we will eliminate the term 'socio-hydrology' since it can be interpreted differently by different researchers which might cause confusion. What we are trying to address is the broad space of human-water interactions and therefore, to make the manuscript clearer, we will replace the term 'socio-hydrology' with 'human-water systems'. Second, we perceive the discussion of results in the context of human-water systems to be important since:  (1) Dynamics of environmental awareness and the rebound effect have been extensively discussed by stakeholders and could be better explained using the concepts of human-water relationships in the existing literature. (2) Stakeholders' conceptualization of human-water relationships were aligned with those mentioned in the literature. Delineating relationships that are pointed out by observations from past studies is valuable to the advancement of the study of such systems. We will emphasize the importance of human-water relationships in the introduction and in the discussion."

14. **RC**: I expect the discussion to be focused on how the proposed framework and the implementation of this case study succeeded (or not) in addressing the limitations of other approaches of participatory modelling, how it is able to address the above mentioned five points and what the limitations are of the framework and methods proposed in this study.

**AR**: Section 5.1 discusses the evaluation of the proposed framework. We will elaborate this part of the discussion and add the limitations and barriers of the implementation of the proposed framework to the section.

15. **RC**: More general, the authors refer to the use of participatory approaches for system dynamics modelling, to me system dynamics modelling suggests the actual translation of the conceptual model in a quantitative version and running the model to check if the outcomes are correct and what is expected. The authors state that the final step of stage 3 is to simulate the policies with the model and discuss this with the stakeholders, however, in my opinion, translating the model into a quantitative version and running simulations should already be done in stage 2, as a check, to make sure the conceptual model makes sense, and model simulations could also help the stakeholder discussions about whether the model accurately represents the situation. I would suggest to include the quantitative model and relevant simulations in the manuscript. If not, I would suggest to rephrase the manuscript to use conceptual models instead of system dynamics models. Since, I think the conceptual loop diagrams that are developed in this paper are conceptual models of the reality but not yet system dynamics models.

**AR**: Thank you for your comment. Although this is an important consideration, the inclusion of a quantitative model in this manuscript is not feasible since it is still a work in progress. We will use the term 'conceptual model' instead of 'system dynamics model' throughout the manuscript.

---

## Author Comment (AC2) · 19 Nov 2020

**Manuscript hess-2020-437**
**Response to Referee #1**

Dear Referee #1:

Thank you for your insightful comments. Your feedback provides valuable improvements to our manuscript. Please find below our responses to your comments.

**Acronyms**
**RC** - Referee comments
**AR** - Author responses

**Comments and responses**

1. **RC**: The manuscript represents a contribution regarding the implementation of participatory modelling in vulnerable and disadvantaged communities to address eutrophication problems. Understanding and applying tools such as participatory system dynamics in different contexts is illuminating for research and practice and therefore, the manuscript offers a valuable work to be published in this special issue of the journal.

   **AR**: Thank you!

2. **RC**: However, the methodology and results associated to the multilevel storylines is really similar compared to the initial phases of conducting traditional participatory modelling processes to elicit causal loop diagrams. In this regard, it is required to strength the comparison of traditional approaches to elicit causal loop diagrams in participatory process and the storylines approach, or otherwise, presenting the storylines in their fair dimensions as an alternative to elicit causal loop diagrams in participatory process.

   **AR**: The approach we suggest is useful as it builds upon CLD construction methods to include more stakeholders meaningfully. We do not perceive it as an entirely new framework but rather as an extension to CLD building that can be implemented within marginalized communities. As pointed in the methodology, our research included iterations between storylines and CLDs. Storylines were used for two purposes:
   - **Extraction of information:** by definition, a storyline describes cause-and-effect relationships between events that impact certain components or actors. Therefore, storylines are compatible with CLDs. The main difference is that storylines provide more leeway for stakeholders to explain their inputs. For example, some stakeholders used metaphors or anecdotes to describe their observations. This is useful in the contexts of (1) less-literate and non-expert stakeholders who (a) might not be able to explicitly place their observations in the context of variables and links and (b) might feel intimidated by the technicalities of the CLD approach, and (2) Indigenous stakeholders who consider storytelling as a way to share knowledge. Although labelled as a 'simplified version of a

storyline', we think that Figure 5 might be oversimplifying and misconstruing the flexibility of storylines. Therefore, we will improve the figure to include the intricacies of an extracted storyline.

- **Dissemination of results and science communication:** disseminating results in the form of storylines is more suitable for an audience of non-experts especially in the context of marginalized communities that include stakeholders who might not be comfortable with deciphering CLDs.

In the context of results, the difference lies in the ability of the methodology to (1) accommodate marginalized stakeholders who might have not been able to effectively participate otherwise and (2) the unique contributions of those stakeholders.

To address your comment, we will highlight the abovementioned points, eliminate terms and phrases that might exaggerate the novelty of the method (e.g. new framework), and emphasize that storylines used in parallel with CLDs allow for more inclusive stakeholder participation.

3. **RC**: The authors reviewed relevant literature regarding the building blocks of the approach adopted. Nonetheless, participatory modelling has been used in water resources management almost from its beginning and some relevant authors regarding the integration of these concepts were overlooked what perhaps lead the authors to not sufficiently acknowledge that practices quite similar to those developed in their research have been employed to: i) construct causal diagrams eliciting the stakeholders´ perspectives; and ii) take part in the whole cycle of system dynamic modeling.

   **AR**: We will add more studies pertaining to participatory SD modelling in water resources management (WRM) to the Background section (e.g. Enteshari et al., 2020; Pagano et al., 2019; Perrone et al., 2020; Stave, 2003; Tidwell et al., 2004). We showed some relevant studies (i.e. CLD building in WRM) that do not explicitly adapt their practices to effectively include marginalized stakeholders who might have been pertinent to corresponding case studies - e.g. Hassanzadeh et al. (2019) did not accommodate a multilingual context; Inam et al. (2015) used methods that require reading and writing.

   We would like to note that what we are looking for is not restricted to participatory methods that aim to elicit different perspectives. We are explicitly seeking approaches that aim to adapt participatory methodologies to include the effective participation of marginalized stakeholders (i.e. methods that are inclusive by design). We will highlight the aforementioned in key parts of the main text.

4. **RC**: The manuscript structure could be substantially improved. There are different ways the manuscript could be better structured. One could be to show a conceptual framework with the original building blocks, followed by a proposal of the integrated approach. Then, the methodology where this integrated approach is materialized could be described. After this, the results could be presented. The current way in which the

document is developed, with parts of the original approaches and the integrated approach appear in the introduction, the background and then the methodology, is not clear. Another path could be to develop the three proposed objectives in the results section that currently only addresses the implementation of integrated approach. In any case, sections 4.1 and 4.2 should not be part of the methodology since those are general descriptions of the approaches and not an account of the activities carried out to undertake the research process.

**AR**: Thank you for your suggestion. This will considerably improve the structure of our manuscript. We will add a section entitled 'Conceptual Framework'. This section will include the building blocks (the storytelling approach and the MLP framework) in addition to the integrated approach. We will remove the Background section. We will move Lines 125-151 and place them in the introduction between lines 93 and 94. Lines 152-235 will be adjusted to fit the section elaborating the storytelling approach in the new Conceptual Framework section.

5. In addition, more insight should be provided regarding the case selection, and how the research idea and problem emerged from the interaction between researchers and stakeholders.

   **AR**: We will elaborate more on that in Section 5.1. In brief, the eutrophication problem in the Lake Atitlan Basin has been a pressing environmental problem for more than a decade. The problem has been (1) especially amplified after cyanobacterial blooms covered 40% of the lake's surface in 2009 (Komárková et al., 2011; Weisman et al., 2018) and the endorsement of the Mega-collector project in 2018 (Esswein and Zernack, 2019) and (2) prioritized by research participants we met with (who were at the time associated with organizations working on projects connected to the lake) (e.g. Centro de Estudios Atitlan).

6. **RC**: Lines 145 -150
   Another approach, stakeholder created causal loop diagrams (CLDs), contain variables connected by links indicating causal relationships. Although CLDs have been previously applied in participatory research (Inam et al., 2015, 2017b), their construction requires reading and writing skills. Hence, they are ill-suited for involving less-literate participants in participatory model- building activities.
   This statement is not necessarily true. It is not necessarily the stakeholders those who formulate the CLDs. This approach of a facilitator building CLDs from interviews or focus groups and "translating" the information provided into the System Dynamics language is widely used in SD in WRM.

   **AR**: Thank you for pointing this out. We will address this by including and discussing studies that translated interviews or focus group discussions to CLDs (e.g. Giordano et al., 2020; Kim and Andersen, 2012; Pham et al., 2020). Also, we will highlight how our storyline methodology is different from researchers simply drawing a CLD based on an

interview. For example, the method applied in this study reduces the researcher's influence on the conceptual model and the number of ambiguous statements usually encompassed by interviews – which are two of the challenges faced when translating interviews to CLDs (Kim and Andersen, 2012).

**RC**: Line 269
Figure 1 Location of the study area in Guatemala. Created in QGIS (https://qgis.org/) using Esri (2009).
The map in Figure 1 needs to be substantially improved. The location of the study area in Guatemala and of Guatemala in America must be shown. Labels relevant only to the study area with Font of an appropriate size together with a grid of coordinates should be included.

**AR**: Thank you for pointing this out. We will revise the figure as per your suggestions.

7. **RC**: Lines 401 - 403

The sign corresponding to each link indicates the type of relationship between the two variables: (+) indicates a direct relationship, while (-) implies an indirect one.
This statement must be reworded. That is not the correct explanation of polarities for the causal relations in System Dynamics.

**AR**: Thank you for pointing this out. We will rephrase the explanation.

8. **RC**: Lines 452 - 455
These policies and BMPs are then simulated in a quantitative version of the model. The results are subsequently presented to stakeholders by members of the guidance team and discussed until an agreement on suitable solutions is reached. This paper does not cover the implementation of this step.
Since a relevant feature of this work is the context of their application, involving marginalized and indigenous communities, it would be an important contribution to explain how model results were discussed and communicated to these stakeholders, which is relevant since the authors expressed that most of them cannot read or write.

**AR**: Thank you for your comment. We will provide a brief explanation. In brief, the disseminated information is synthesized in a comprehensive narrative and communicated using storylines that appeal to non-expert audiences.

9. **RC**: Line 474
Table 1: Demographics of project participants
It would be convenient to expand the information on the number of participating indigenous communities and their different languages

**AR**: We will expand the table to include the number of participating Kaqchikel, Tz'utujil, and K'iche stakeholders.

10. **RC**: Lines 511 – 512

From the Macro-level storylines elicited from primary researcher participants, the authors concluded that the model should address the eutrophication problem of Lake Atitlán. According to that statement ¿how this research fits within a real participatory approach in which external agents (researchers) should have a facilitating role in a process through which the relevant stakeholders reflect, deliberate and are empowered to make decisions, rather than a role of extracting information from stakeholders and make decisions for them?

**AR**: Thank you for your comment. We agree that the term 'conclude' indicates that the researchers have made a unilateral decision. Therefore, we will rephrase this sentence.

- As mentioned in point 5 above, the problem has been prioritized by researcher participants who themselves have been involved in projects associated with the lake's eutrophication.
- The goal of the activity was to support community-based decision-making, especially by allowing stakeholders to communicate their perspectives, needs, and priorities.
- Members of the guidance team were aware that this presented a learning opportunity for them as well and sought to remain cognizant of their positionality in the research setting.

We will highlight the aforementioned points in the text.

11. **RC**: Figure 7

Assess whether there is a feedback loop between "crop productivity" and "use of inorganic fertilizers and pesticides"
Improve the figure so that the polarity between "irrigation efficiency" and "untreated wastewater can be observed".

**AR**: Thank you for pointing this out. After we assessed the interactions between crop productivity and the consumption of inorganic fertilizers and pesticides, we have found that a feedback loop between the two variables exists. In general, the increased use of inorganic fertilizers and pesticides increases crop productivity (De Ponti et al., 2012). However, this was not mentioned by stakeholders and therefore, we will be consulting agriculturists/farmers who participated in the activity to validate the addition of the feedback loop. We will add relevant justification to the text.

We believe you meant "water shortage" instead of untreated wastewater. If so, we will revise as suggested.

12. **RC:** Figure 8

Improve the figure so that the polarity between "WWTP" and "untreated wastewater" can be observed.
Use the term in full for WWTP
Improve the figure so that the polarity between "available land" and "septic tanks" can be observed.
correct typo in septic tanks

Correct the polarity between Jobs and Poverty. This polarity should be negative, not positive

**AR**: Thank you for pointing this out. We will revise the figure as per your suggestions.

13. **RC:** Figure 11
Is confusing that a relation can be reinforcing and balancing at the same time. Please clarify.

**AR**: Figure 11 displays a **generalized relationship** between economic prosperity and nutrient enrichment. The CLD on the right shows 2 loops: 1 balancing and 1 reinforcing. The figure is used to highlight feedback loops.

For both loops, and as explained in the Consequences section (Lines 555-565), the causal link corresponding to the impact of nutrient enrichment on economic prosperity is negative. This causal link is generalized and does not contain intermediaries since the point of the figure is to elaborate on the feedback (i.e. the impact of economic prosperity on nutrient enrichment).

Figure 11 (a): Some stakeholders stated that economic prosperity increases potential investments in WWTPs which reduces the discharge of untreated wastewater, consequently decreasing nutrient enrichment. This decrease in nutrient enrichment would lead to an increase in economic prosperity. The causal link corresponding to the impact of economic prosperity on nutrient enrichment is negative. Therefore, the relationship between economic prosperity and nutrient enrichment in this case is represented by a reinforcing loop (Fig. 11 (a)).

Other participants implied that economic prosperity increases investments in tourism businesses, which increases the number of tourists, consequently increasing the amount of untreated wastewater. This leads to an increase in nutrient enrichment which would cause a decrease in economic prosperity. The causal link corresponding to the impact of economic prosperity on nutrient enrichment is positive. Therefore, the relationship between economic prosperity and nutrient enrichment in this case is represented by a balancing loop (Fig. 11 (b)).

Both processes were the result of the inclusive participatory process and show the added value of incorporating marginalized stakeholders since the balancing loop between the two variables was exclusively identified by Indigenous stakeholders. Additionally, the delineation of both relationships shows that all potentially valid points can be represented explicitly in the model (which reinforces the point of inclusivity). However, we acknowledge that one of the two loops will dominate model behaviour. This will depend on model quantification.

To make the figure clearer and less confusing we will:

- Add intermediaries to the causal link corresponding to the impact of economic prosperity on nutrient enrichment
- Provide a clearer explanation in the caption
- Replace the current example (the loops on the right) with two examples mentioned within lines 570-587 and refer to the figure right next to the examples it represents
- Emphasize that the figure represents a generalized relationship
- Mention that model quantification will show which of the two loops will dominate model behaviour

**References**

Enteshari, S., Safavi, H. R. and van der Zaag, P.: Simulating the interactions between the water and the socio-economic system in a stressed endorheic basin, Hydrol. Sci. J., 65(13), 2159–2174, doi:10.1080/02626667.2020.1802027, 2020.

Esswein, A. and Zernack, F.: El historíco proyecto que quiere salvar el lago Atitlán pero no gusta a todos, El País, 2019.

Giordano, R., Pluchinotta, I., Pagano, A., Scrieciu, A. and Nanu, F.: Enhancing nature-based solutions acceptance through stakeholders' engagement in co-benefits identification and trade-offs analysis, Sci. Total Environ., 713, 136552, doi:10.1016/j.scitotenv.2020.136552, 2020.

Kim, H. and Andersen, D. F.: Building confidence in causal maps generated from purposive text data: Mapping transcripts of the Federal Reserve, Syst. Dyn. Rev., 28(4), 311–328, doi:10.1002/sdr.1480, 2012.

Komárková, J., Dix, M., Komárek, J., Girón, N. and Rejmánková, E.: Cyanobacterial blooms in Lake Atitlan, Guatemala, Limnologica, 41(4), 296–302, doi:10.1016/j.limno.2010.12.003, 2011.

Pagano, A., Pluchinotta, I., Pengal, P., Cokan, B. and Giordano, R.: Engaging stakeholders in the assessment of NBS effectiveness in flood risk reduction: A participatory System Dynamics Model for benefits and co-benefits evaluation, Sci. Total Environ., 690, 543–555, doi:10.1016/j.scitotenv.2019.07.059, 2019.

Perrone, A., Inam, A., Albano, R., Adamowski, J. and Sole, A.: A participatory system dynamics modeling approach to facilitate collaborative flood risk management: A case study in the Bradano River (Italy), J. Hydrol., 580(November 2019), doi:10.1016/j.jhydrol.2019.124354, 2020.

Pham, Y., Reardon-Smith, K., Mushtaq, S. and Deo, R.: Feedback modelling of the impacts of drought: a case study in coffee production systems in Viet Nam, Clim. Risk Manag., 30(February), 100255, doi:10.1016/j.crm.2020.100255, 2020.

De Ponti, T., Rijk, B. and Van Ittersum, M. K.: The crop yield gap between organic and conventional agriculture, Agric. Syst., 108, 1–9, doi:10.1016/j.agsy.2011.12.004, 2012.

Stave, K. A.: A system dynamics model to facilitate public understanding of water management options in Las Vegas, Nevada, J. Environ. Manage., 67(4), 303–313, doi:10.1016/S0301-4797(02)00205-0, 2003.

Tidwell, V. C., Passell, H. D., Conrad, S. H. and Thomas, R. P.: System dynamics modeling for community-based water planning: Application to the Middle Rio Grande, Aquat. Sci., 66(4), 357–372, doi:10.1007/s00027-004-0722-9, 2004.

Weisman, A., Chandra, S., Rejmánková, E. and Carlson, E.: Effects of Nutrient Limitations and Watershed Inputs on Community Respiration in a Deep, Tropical Lake: Comparison of Pelagic and Littoral Habitats, Water Resour. Res., 54(8), 5213–5224, doi:10.1029/2017WR021981, 2018.

---

## Author Response (AR1)

**Manuscript hess-2020-437**
**Response to Editor**

Dear Editor:

We are grateful for your valuable comments. Your feedback provides valuable insight to our manuscript.

**Acronyms**
**EC** - Editor's comments
**AR** - Author responses

**Comments and responses**

1.  **EC**:  Consider removing "socio-hydrological" from the title to reflect the wording of the revised manuscript.

    **AR**: The title has been revised accordingly.

2.  **EC**: The concept of using these approaches to reveal leverage points is intriguing. The notion of leverage points you use in this paper, is that some interventions can be expected to have much greater impacts than other interventions. Regarding those that you identify in section 5.3, can you give some indication of which ones might have the greatest impact on the system? Can you highlight any leverage points that were counterintuitive in terms of priorities or strategies identified by the stakeholders versus their potential impacts on water quality?

    **AR**: Thank you for your comment. We added some insights on proposed solutions and leverage points (Lines 883-900):

    "Quantification is needed to assess the impacts of suggested solutions. Nevertheless, some insights can be identified from this activity. For example, wastewater treatment, which was discussed by stakeholders, could play an important role in decreasing the discharge of untreated wastewater produced by residents and tourists (R6 in Fig.9). However, about 60% of wastewater in the area is not discharged through a drainage system. Therefore, on the contrary to what some stakeholders suggested, the proposed plan would not present an optimum solution unless coupled with other projects such as drainage system planning and dry toilets. On another note, aiming to reduce the consumption of inorganic fertilizers by supporting organic agriculture (as mentioned by participants in Sect. 5.3) could potentially decrease the contribution of agricultural activities to nutrient enrichment (Fig.8). In this light, subsidies on inorganic fertilizers present an interesting leverage point in the system. Reexamining subsidies and reallocating financial resources to incentivize organic agriculture might play a role in increasing the efficiency of fertilizer application and, consequently, decrease nutrient enrichment in the lake. Finally, the goal of the system is a potent leverage point (Fischer

and Riechers, 2019; Meadows, 1999). In this case, rethinking the goal, which focuses on decreasing nutrient enrichment, might be useful. This was not explicitly mentioned by stakeholders as a solution but rather implicitly through discussions about the 'Mega-collector'. The 'Mega-collector' project was opposed by many stakeholders partially since they anticipate that, while addressing the lake's water quality problems, it could also lead to other problems (e.g. water shortage, economic disparities and loss of biodiversity). Therefore, shifting the goal of the system to focus on an environmental component that could offer a more holistic view of the system's wellbeing, such as biodiversity, might be useful."

3. **EC**:  Some font in Fig 2 is small and should be enlarged. The font in Figure 7 is also very small and should be enlarged.

   **AR**: Fig. 2 has been adjusted accordingly. We were not able to enlarge the font within shapes in Fig.7. However, we provided a larger figure.

**Manuscript hess-2020-437**
**Response to Referee #1**

Dear Referee #1:

Thank you for your insightful comments. Your feedback provides valuable improvements to our manuscript. Please find below our responses to your comments.

**Acronyms**
**RC** - Referee comments
**AR** - Author responses

**Comments and responses**

1. **RC**: The manuscript represents a contribution regarding the implementation of participatory modelling in vulnerable and disadvantaged communities to address eutrophication problems. Understanding and applying tools such as participatory system dynamics in different contexts is illuminating for research and practice and therefore, the manuscript offers a valuable work to be published in this special issue of the journal.

   **AR**: Thank you!

2. **RC**: However, the methodology and results associated to the multilevel storylines is really similar compared to the initial phases of conducting traditional participatory modelling processes to elicit causal loop diagrams. In this regard, it is required to strength the comparison of traditional approaches to elicit causal loop diagrams in participatory process and the storylines approach, or otherwise, presenting the storylines in their fair dimensions as an alternative to elicit causal loop diagrams in participatory process.

   **AR**: The approach we suggest is useful as it builds upon CLD construction methods to include more stakeholders meaningfully. We do not perceive it as an entirely new framework but rather as an extension to CLD building that can be implemented within marginalized communities. As pointed in the methodology, our research included iterations between storylines and CLDs. Storylines were used for two purposes:
   - **Extraction of information:** by definition, a storyline describes cause-and-effect relationships between events that impact certain components or actors. Therefore, storylines are compatible with CLDs. The main difference is that storylines provide more leeway for stakeholders to explain their inputs. For example, some stakeholders used metaphors or anecdotes to describe their observations. This is useful in the contexts of (1) less-literate and non-expert stakeholders who (a) might not be able to explicitly place their observations in the context of variables and links and (b) might feel intimidated by the technicalities of the CLD approach, and (2) Indigenous stakeholders who consider storytelling as a way to share knowledge.

- **Dissemination of results and science communication:** disseminating results in the form of storylines is more suitable for an audience of non-experts especially in the context of marginalized communities that include stakeholders who might not be comfortable with deciphering CLDs.

To address your comment, we highlighted the abovementioned points (Lines 351-362), eliminated terms and phrases that might exaggerate the novelty of the method and emphasized that storylines used in parallel with CLDs allow for more inclusive stakeholder participation (Lines 137-139 and Lines 352-353). We also improved Figure 5 to better match the wording of simplified storylines.

3. **RC**: The authors reviewed relevant literature regarding the building blocks of the approach adopted. Nonetheless, participatory modelling has been used in water resources management almost from its beginning and some relevant authors regarding the integration of these concepts were overlooked what perhaps lead the authors to not sufficiently acknowledge that practices quite similar to those developed in their research have been employed to: i) construct causal diagrams eliciting the stakeholders´ perspectives; and ii) take part in the whole cycle of system dynamic modeling.

**AR**: We added more studies pertaining to participatory causal loop diagrams in water resources management – Lines 122 to134 (e.g. Enteshari et al., 2020; Pagano et al., 2019; Perrone et al., 2020; Stave, 2003; Tidwell et al., 2004). As per the suggestion of Referee #2, we tried to reduce the use of the term SD modelling and therefore addressed CLDs instead.

4. **RC**: The manuscript structure could be substantially improved. There are different ways the manuscript could be better structured. One could be to show a conceptual framework with the original building blocks, followed by a proposal of the integrated approach. Then, the methodology where this integrated approach is materialized could be described. After this, the results could be presented. The current way in which the document is developed, with parts of the original approaches and the integrated approach appear in the introduction, the background and then the methodology, is not clear. Another path could be to develop the three proposed objectives in the results section that currently only addresses the implementation of integrated approach. In any case, sections 4.1 and 4.2 should not be part of the methodology since those are general descriptions of the approaches and not an account of the activities carried out to undertake the research process.

**AR**: We added a section entitled 'Conceptual Framework'. This section includes the building blocks (the storytelling approach and the MLP framework) in addition to the integrated approach. We removed the Background section. Previously Lines 125-151 (currently Lines 99-121) were placed in the introduction.

5.  In addition, more insight should be provided regarding the case selection, and how the research idea and problem emerged from the interaction between researchers and stakeholders.

    **AR**: We added some information to Section 5.1 (Lines 635-645). In brief, the eutrophication problem in the Lake Atitlan Basin has been a pressing environmental problem for more than a decade. The problem has been (1) especially amplified after cyanobacterial blooms covered 40% of the lake's surface in 2009 (Komárková et al., 2011; Weisman et al., 2018) and the endorsement of the Mega-collector project in 2018 (Esswein and Zernack, 2019) and (2) prioritized by research participants we met with (who were at the time associated with organizations working on projects connected to the lake).

6.  **RC**: Lines 145 -150
    Another approach, stakeholder created causal loop diagrams (CLDs), contain variables connected by links indicating causal relationships. Although CLDs have been previously applied in participatory research (Inam et al., 2015, 2017b), their construction requires reading and writing skills. Hence, they are ill-suited for involving less-literate participants in participatory model- building activities.
    This statement is not necessarily true. It is not necessarily the stakeholders those who formulate the CLDs. This approach of a facilitator building CLDs from interviews or focus groups and "translating" the information provided into the System Dynamics language is widely used in SD in WRM.

    **AR**: Thank you for pointing this out. We addressed this in Lines 351-363 and Lines 127-133 by including and discussing studies that translated interviews or focus group discussions to CLDs (e.g. Giordano et al., 2020; Kim and Andersen, 2012; Pham et al., 2020). We highlighted how the multi-level storyline methodology is different from researchers simply drawing a CLD based on an interview: "This method presents two main issues: (1) it increases the risk of researchers' influences on the model and (2) it might yield ambiguous statements, prone to misinterpretation (Kim and Andersen, 2012). Both are especially critical in the context of marginalized communities, where perspectives of less-powerful stakeholders tend to be lost or disregarded (Butler and Adamowski, 2015; Cooke and Kothari, 2001)."

**RC**: Line 269
    Figure 1 Location of the study area in Guatemala. Created in QGIS (https://qgis.org/) using Esri (2009).
    The map in Figure 1 needs to be substantially improved. The location of the study area in Guatemala and of Guatemala in America must be shown. Labels relevant only to the study area with Font of an appropriate size together with a grid of coordinates should be included.

    **AR**: Revised (NB: The Figure's number has been changed from 1 to 2)

7. **RC**: Lines 401 - 403

   The sign corresponding to each link indicates the type of relationship between the two variables: (+) indicates a direct relationship, while (-) implies an indirect one.
   This statement must be reworded. That is not the correct explanation of polarities for the causal relations in System Dynamics.

   **AR**: Corrected (Lines 545-550)

8. **RC**: Lines 452 - 455
   These policies and BMPs are then simulated in a quantitative version of the model. The results are subsequently presented to stakeholders by members of the guidance team and discussed until an agreement on suitable solutions is reached. This paper does not cover the implementation of this step.
   Since a relevant feature of this work is the context of their application, involving marginalized and indigenous communities, it would be an important contribution to explain how model results were discussed and communicated to these stakeholders, which is relevant since the authors expressed that most of them cannot read or write.

   **AR**: We provided a brief explanation (Lines 603-604).

9. **RC**: Line 474
   Table 1: Demographics of project participants
   It would be convenient to expand the information on the number of participating indigenous communities and their different languages

   **AR**: Added (Table 1)

10. **RC**: Lines 511 – 512
    From the Macro-level storylines elicited from primary researcher participants, the authors concluded that the model should address the eutrophication problem of Lake Atitlán. According to that statement ¿how this research fits within a real participatory approach in which external agents (researchers) should have a facilitating role in a process through which the relevant stakeholders reflect, deliberate and are empowered to make decisions, rather than a role of extracting information from stakeholders and make decisions for them?

**AR**: We rephrased this sentence (Lines 675-676).
- As mentioned in point 5 above, the problem has been prioritized by researcher participants who themselves have been involved in projects associated with the lake's eutrophication.
- The goal of the activity was to support community-based decision-making, especially by allowing stakeholders to communicate their perspectives, needs, and priorities.

- Members of the guidance team were aware that this presented a learning opportunity for them as well and sought to remain cognizant of their positionality in the research setting.

We highlighted the aforementioned points in the text (Lines 617 – 622).

11. **RC**: Figure 7
Assess whether there is a feedback loop between "crop productivity" and "use of inorganic fertilizers and pesticides"
Improve the figure so that the polarity between "irrigation efficiency" and "untreated wastewater can be observed".

**AR**:  We started the consultation process with agriculturists/farmers who participated in the activity to validate the addition of the feedback loop. However, due to time constraints and COVID19 restrictions, a conclusion about the existence and the nature of the feedback loop has not been made. Therefore, we added this as a limitation (Lines 910-912).

We believe you meant "water shortage" instead of untreated wastewater. If so, we revised as suggested.

12. **RC:** Figure 8
Improve the figure so that the polarity between "WWTP" and "untreated wastewater" can be observed.
Use the term in full for WWTP
Improve the figure so that the polarity between "available land" and "septic tanks" can be observed.
correct typo in septic tanks
Correct the polarity between Jobs and Poverty. This polarity should be negative, not positive

**AR**: Revised

13. **RC:** Figure 11
Is confusing that a relation can be reinforcing and balancing at the same time. Please clarify.

**AR**: Figure 11 displays a **generalized relationship** between economic prosperity and nutrient enrichment. The generalization has been agreed upon by participants. The CLD on the right shows 2 loops: 1 balancing and 1 reinforcing. The figure is used to highlight feedback loops.

Both processes were the result of the inclusive participatory process and show the added value of incorporating marginalized stakeholders since the balancing loop between the two variables was exclusively identified by Indigenous stakeholders. Additionally, the delineation of both relationships shows that all potentially valid points

can be represented explicitly in the model (which reinforces the point of inclusivity). However, we acknowledge that one of the two loops will dominate model behaviour. This will depend on model quantification.

To make the figure clearer and less confusing we:
- Provided a clearer explanation in the caption (Lines 650-655): "Figure 11: The relationship between nutrient enrichment in Lake Atitlán and economic prosperity (reinforcing loop (a) and balancing loop (b)). The two loops on the left represent generalized relationships of the two loops on the right, mentioned and agreed upon by participants. The two contradicting views underpinning the two generalized relationships (loops (a) and (b) on the left) were elicited by different stakeholder groups. The delineation of both relationships shows that all potentially valid points can be represented explicitly in the model, which reinforces the point of inclusivity. Quantification would show which of the two loops dominates the model's behaviour."
- Replaced the loops on the right with loops extracted from Fig. 9.
- Emphasized that this is a generalized relationship described and agreed upon by stakeholders (Lines 734-735).

**Manuscript hess-2020-437**
**Response to Referee #2**

Dear Referee #2:

We are grateful for your valuable comments. Your feedback will help us improve our manuscript significantly. Please find below our responses to your comments.

**Acronyms**
**RC** - Referee comments
**AR** - Author responses

**Comments and responses**

1.  **RC**: The authors present a method for participatory modelling for system dynamics models. The topic is interesting and potentially a nice contribution to the existing literature, but I do think the manuscript needs significant improvements.

    **AR**: Thank you! We appreciate your constructive feedback that provides valuable improvements to our manuscript.

2.  **RC**: The authors provide a nice overview of participatory approaches and the limitations of current methods, however, to me it does not become entirely clear how the method applied in this research is different from already existing methods. It seems like the main difference is the fact that the authors used indigenous languages for conducting their interviews, which does not really make it a new method.

    **AR**: The approach we suggest is useful as it builds upon CLD construction methods to include more stakeholders meaningfully. We do not perceive it as an entirely new framework but rather as an extension to CLD building that can be implemented within marginalized communities. As pointed in the methodology, our research included iterations between storylines and CLDs. Storylines were used for two purposes:

    - **Extraction of information:** by definition, a storyline describes cause-and-effect relationships between events that impact certain components or actors. Therefore, storylines are compatible with CLDs. The main difference is that storylines provide more leeway for stakeholders to explain their inputs. For example, some stakeholders used metaphors or anecdotes to describe their observations. This is useful in the contexts of (1) less-literate and non-expert stakeholders who (a) might not be able to explicitly place their observations in the context of variables and links and (b) might feel intimidated by the technicalities of the CLD approach, and (2) Indigenous stakeholders who consider storytelling as a way to share knowledge.

- **Dissemination of results and science communication:** disseminating results in the form of storylines is more suitable for an audience of non-experts especially in the context of marginalized communities that include stakeholders who might not be comfortable with deciphering CLDs.

To address your comment, we highlighted the abovementioned points (Lines 351-362), eliminated terms and phrases that might exaggerate the novelty of the method and emphasized that storylines used in parallel with CLDs allow for more inclusive stakeholder participation (Lines 137-139 and Lines 352-353). We also improved Figure 5 to better match the wording of simplified storylines.

3. **RC**: According to the authors, the new framework should be able to "(1) incorporate effective participation of marginalized stakeholders, (2) induce collaboration, (3) integrate diverse perspectives, (4) facilitate model conceptualization and (5) produce description of relevant socio-hydrological phenomena."

   **AR**: Our responses to each of these points are found below: points 4 to 7.

4. **RC**: Point 1 is addressed partly by using the native language of participants but I would expect that this would also be addressed in the stakeholder selection process, for example, how do you ensure that these marginalized stakeholders are not left out? The process described in section 4.3 states that in stage 1 one starts with developing a focus group with primary stakeholders, how did the researchers make sure these stakeholders were representative?

   **AR**: Thank you for your comment. First, primary stakeholders (or researcher participants) included Indigenous stakeholders. Second, the guidance team (made up of three individuals in total) also included an Indigenous researcher from Tz'oloj Ya', associated with Universidad Rafael Landívar. Third, during the focus group discussion, the guidance team explicitly addressed the socio-cultural dimensions of the Lake Atitlan Basin and included stakeholders that represent those dimensions. Fourth, the guidance team was actively seeking Indigenous stakeholders and institutions (including traditional councils and youth groups). We emphasize the aforementioned points in the text:
   - Lines 482-484: "It is important to select researcher participants from different professional and social backgrounds and who identify as belonging to marginalized groups, to construct a holistic view of the problem."
   - Line 496: "They also ensure that marginalized communities are discussed."
   - Lines 506-509: "The guidance team explicitly delineates stakeholders representing the different dimensions (economic, social, cultural, and political), mentioned in the Macro-level storyline. The team actively seeks individuals and organizations that are associated with marginalized communities."
   - Lines 636-637:"Researcher participants included individuals who identify as belonging to marginalized groups."

5. **RC**: Point 2 is not really addressed in the rest of the manuscript. How do the authors ensure that the proposed framework induced collaboration? Did this work? Did collaboration increase after the participatory modelling exercise?

   **AR**: Thank you! We agree. We changed the wording from 'collaboration' to 'dialogue'.. The participatory activity allowed different stakeholder groups, Indigenous and Hispanic stakeholders to discuss, propose, and share solutions in the two workshops. Establishing a sense of trust with the Indigenous community and gaining their confidence required the implementation of a process that was truly tailored to those communities. Multiple stakeholders stated that they had lost confidence in such processes since previous participatory approaches in the area did not effectively incorporate them. As stated in Lines 658-860: "Instead of effectively integrating Indigenous communities in decision-making, previously conducted participatory processes often reinforced illegitimate and unjust decisions, while claiming them as 'participatory.'" Hence, the willingness of some indigenous stakeholders to start a dialogue and communicate with other stakeholder groups towards finding solutions was triggered by the process which is inclusive by design and was conducted in a culturally relevant way. We highlighted the aforementioned in the text (Lines 862-864): "This trust triggered the willingness of some Indigenous participants to start a dialogue and communicate with other stakeholder groups. Carried out in a culturally relevant way, the participatory process allowed Indigenous and Hispanic stakeholders to discuss and share solutions during workshops."

6. **RC**: Point 3 is discussed a bit more, in the sense that the storylines of different stakeholders allow for different perspectives, but it is not clear how the different perspectives are integrated into one conceptual model and how in this process it is ensured that the views of marginalized stakeholders do not get lost.

   **AR**: The merged storyline and CLD contains all variables and relationships extracted from stakeholders.We emphasized that all views are conserved:
   - Lines 357-363:"Moreover, the method is explicitly and systemically designed to dynamically translate from storylines to CLDs and vice versa, which makes (1) stakeholders' statements less prone to misinterpretation and (2) the process less susceptible to researchers' influences, compared to other CLD-building processes that require ex-post extraction of CLDs from interviews or focus group discussions (Giordano et al., 2020; Pham et al., 2020). This facilitates the conservation of stakeholders' views."
   - Lines 553-554: "The guidance team ensures that all views are conserved and included in each individual CLD."
   - Lines 563-564:"Ensuring the conservation of all identified relationships, each individual CLD is joined, forming an overall merged CLD - as per Inam et al. (2015)."
   - Line 869-870: "All these contributions were conserved and included in the conceptual model."

As shown in Table 2, unique contributions of different stakeholder groups were pointed out, discussed, and included in the model. To emphasize this point, we highlighted the table in the text (Lines 868-869).

7. **RC**: Point 5 is only discussed at the end of the manuscript in the discussion and it is not clear from the start what socio-hydrological phenomena are and why it is important that the participatory modelling process produces descriptions of phenomena.

   **AR**: To respond to this comment, we reiterate our response to point 7 above:
   First, we would like to point out that we eliminated the term 'socio-hydrology' since it can be interpreted differently by different researchers which might cause confusion. What we are trying to address is the broad space of human-water interactions and therefore, to make the manuscript clearer, we replaced the term 'socio-hydrology' with 'human-water systems'. Second, we perceive the discussion of results in the context of human-water systems to be important since:  (1) Dynamics of environmental awareness and the rebound effect have been extensively discussed by stakeholders and could be better explained using the concepts of human-water relationships in the existing literature. (2) Stakeholders' conceptualization of human-water feedbacks were aligned with those mentioned in the literature. Delineating relationships that are pointed out by observations from past studies is valuable to the advancement of the study of such systems. We emphasized the importance of human-water feedbacks in the:
   - Introduction:The identification of such feedbacks is important to better inform and conceptualize human-water systems. (Lines 61-63)
   - Discussion: Dynamics of human-water feedbacks discussed by stakeholders were aligned with those mentioned in the literature.: the Rebound Effect (Dumont et al., 2013) and the Pendulum Swing (Van Emmerik et al., 2014). This shows how storytelling is compatible with human-water systems; it facilitated the capture of abstract concepts encompassed by human-water feedbacks that might not have been identified using other model-building methods or data sources. The identification of relationships that have been observed or pointed out by previous studies is valuable to the advancement of the study of human-water systems. (Lines 922 - 929)

8. **RC**: The results section is very unclear to me. First of all the authors should perhaps check the system dynamics literature again for a clear description of a causal loop diagram. The authors mention that a plus indicates a direct relationship and a minus an indirect relationships. In system dynamics a plus usually denotes a positive causal relationship (i.e. if the influencing variable increases the influenced variable also increases) and a minus a negative causal relationship (i.e. if the influencing variable increases the influenced variable decreases).

   **AR**: Thank you for pointing this out. We reworded accordingly (Lines 544-550).

9. **RC**: Also the authors' description of feedback loops is a bit confusing. In system dynamics the feedback loops are the loops that are indicated with B1, R1, etc. However, the authors seem to reference another feedback loop that consists of multiple feedback loops, it is not very clear what this means.

   **AR**: We changed Figure 4 to include 1 balancing loop and 1 reinforcing loop.

10. **RC**: In general it is not clear to me how the causal loop diagrams in Figure 8, 9 and 10 are related to each other. Are they submodels of the main conceptual model? In that case, it would be good to explain how the different submodels are connected. Or are they three different models that each provide a potential explanation for the model, based on different storylines?

    **AR**: They are submodules of one conceptual model. We provided the complete conceptual model as supplementary material.

11. **RC**: Also, the results section does not describe very clearly what the different storylines are that came out of the participatory process and how they were integrated and translated into these conceptual models.

    **AR**: Thank you for your comment. The results section summarizes findings from the storylines. We emphasized that in the text (Lines 619-622). As pointed out in the methodology section:

    (1) Macro-level storylines set the context of the conceptual model by (a) informing the stakeholder analysis and (b) providing interviewed stakeholders with background information that contextualize Meso-level storylines.

    (2) Informed and contextualized by the Macro-level storyline, Meso-level storylines were extracted from stakeholders. Afterwards, each storyline was translated to a CLD. Individual CLDs were merged (forming a merged CLD) and then translated to a merged storyline for the purpose of dissemination - i.e. communicating the results with marginalized stakeholders (figures 5 and 7). The process was iterative until consensus on the merged storyline was reached (Figure 6).

    (3) The sub-modules in figures 8,9, and 10 are the result of steps (1) and (2).

    (4) Micro-level storylines were extracted to provide potential solutions to weak zones exposed by the conceptual model (communicated to less-literate stakeholders using storytelling). They were not incorporated in the current conceptual model since scenario simulation is outside the scope of the study.

12. **RC**: The function of Figure 11 is not clear to me. Is this a simplified version of the conceptual model? Did all the stakeholders agree to this simplified version?

**AR**: Figure 11 displays a **generalized relationship** between economic prosperity and nutrient enrichment. The generalization has been agreed upon by participants. The CLD on the right shows 2 loops: 1 balancing and 1 reinforcing. The figure is used to highlight feedback loops.

Both processes were the result of the inclusive participatory process and show the added value of incorporating marginalized stakeholders since the balancing loop between the two variables was exclusively identified by Indigenous stakeholders. Additionally, the delineation of both relationships shows that all potentially valid points can be represented explicitly in the model (which reinforces the point of inclusivity). However, we acknowledge that one of the two loops will dominate model behaviour. This will depend on model quantification.

To make the figure clearer and less confusing we:

- Provided a clearer explanation in the caption (Lines 650-655): "Figure 11: The relationship between nutrient enrichment in Lake Atitlán and economic prosperity (reinforcing loop (a) and balancing loop (b)). The two loops on the left represent generalized relationships of the two loops on the right, mentioned and agreed upon by participants. The two contradicting views underpinning the two generalized relationships (loops (a) and (b) on the left) were elicited by different stakeholder groups. The delineation of both relationships shows that all potentially valid points can be represented explicitly in the model, which reinforces the point of inclusivity. Quantification would show which of the two loops dominates the model's behaviour."
- Replaced the loops on the right with loops extracted from Fig. 9.
- Emphasized that this is a generalized relationship described and agreed upon by stakeholders (Lines 734-735).

13. **RC**: In the discussion the authors discuss two socio-hydrological phenomena that are relevant for the case study. This is a bit disconnected from the rest of the study. Why is this relevant? And if it is relevant, this should be discussed earlier on in the paper.

    **AR**: To respond to this comment, we reiterate our response to point 7 above:
    First, we would like to point out that we eliminated the term 'socio-hydrology' since it can be interpreted differently by different researchers which might cause confusion. What we are trying to address is the broad space of human-water interactions and therefore, to make the manuscript clearer, we replaced the term 'socio-hydrology' with 'human-water systems'. Second, we perceive the discussion of results in the context of human-water systems to be important since: (1) Dynamics of environmental awareness and the rebound effect have been extensively discussed by stakeholders and could be better explained using the concepts of human-water relationships in the existing literature. (2) Stakeholders' conceptualization of human-water feedbacks were aligned with those

mentioned in the literature. Delineating relationships that are pointed out by observations from past studies is valuable to the advancement of the study of such systems. We emphasized the importance of human-water feedbacks in the:

- Introduction:The identification of such feedbacks is important to better inform and conceptualize human-water systems. (Lines 61-63)
- Discussion: Dynamics of human-water feedbacks discussed by stakeholders were aligned with those mentioned in the literature.: the Rebound Effect (Dumont et al., 2013) and the Pendulum Swing (Van Emmerik et al., 2014). This shows how storytelling is compatible with human-water systems; it facilitated the capture of abstract concepts encompassed by human-water feedbacks that might not have been identified using other model-building methods or data sources. The identification of relationships that have been observed or pointed out by previous studies is valuable to the advancement of the study of human-water systems. (Lines 922 - 929)

14. **RC**: I expect the discussion to be focused on how the proposed framework and the implementation of this case study succeeded (or not) in addressing the limitations of other approaches of participatory modelling, how it is able to address the above mentioned five points and what the limitations are of the framework and methods proposed in this study.

    **AR**: Section 5.1 discusses the evaluation of the proposed framework. We added the limitations of the study to the section (Lines 902-912).

15. **RC**: More general, the authors refer to the use of participatory approaches for system dynamics modelling, to me system dynamics modelling suggests the actual translation of the conceptual model in a quantitative version and running the model to check if the outcomes are correct and what is expected. The authors state that the final step of stage 3 is to simulate the policies with the model and discuss this with the stakeholders, however, in my opinion, translating the model into a quantitative version and running simulations should already be done in stage 2, as a check, to make sure the conceptual model makes sense, and model simulations could also help the stakeholder discussions about whether the model accurately represents the situation. I would suggest to include the quantitative model and relevant simulations in the manuscript. If not, I would suggest to rephrase the manuscript to use conceptual models instead of system dynamics models. Since, I think the conceptual loop diagrams that are developed in this paper are conceptual models of the reality but not yet system dynamics models.

    **AR**: Thank you for your comment. Although this is an important consideration, the inclusion of a quantitative model in this manuscript is not feasible since it is still a work in progress. We used the terms 'conceptual model' and 'systems thinking' (a term used by John Sterman (2000) to describe the process of conceptualizing a systems model) instead of 'system dynamics model' throughout the manuscript. However, we used the

term 'systems model' when referring to an actual SD model. This referral was necessary at times since the goal of the method is to conceptualize systems models exclusively.

---

## Author Response (AR2)

**Manuscript hess-2020-437**
**Response to Editor**

Dear Editor:

Thank you for your feedback.

Editor's comment: The authors have addressed the reviewers' and my own comments very comprehensively to produce an extremely interesting and descriptive manuscript. My only outstanding concern is that Figure 2, location of the study area, needs improvement.

Authors' response: We have fixed the Figure accordingly.

Thanks!